# CO$_2$ subsurface mineral storage by its co-injection with recirculating water

Eric H. Oelkers[1,2 ✉], Serguey Arkadakskiy[3], Zeyad Ahmed[3], Noushad Kunnummal[3], Jakub Fedorik[3], Massimo Marchesi[4], Mouadh Addassi[2], Abdirizak Omar[2], Niccolo Menegoni[2], Sigurdur R. Gislason[1], Grimur Bjornsson[5], Davide Berno[2], Thomas Finkbeiner[2], Abdulkader Afifi[2] & Hussein Hoteit[2]

Carbon capture and storage (CCS) has the potential to help nations meet their Paris Agreement CO$_2$ reduction commitments[1,2]. The ability to capture CO$_2$ within mafic and ultramafic rocks through mineralization of carbon is an example of such a CCS technology[3,4], but large-scale deployment has yet to be achieved[5,6]. Each geologic environment in the Earth's crust requires a distinct carbon storage solution. Whereas some regions of the subsurface contain saline aquifers and sedimentary traps suitable for traditional carbon storage through the injection of high-pressure, dense CO$_2$ below impermeable caprocks, other regions may lack caprocks[5–9]. In these regions, carbon storage is possible through the mineralization of injected water-dissolved CO$_2$ forming stable carbonate minerals through its reactions with reactive silicate rocks and minerals[6,10,11]. A notable challenge to applying this process at scale is that it can require 20–50 times or more water than the mass of CO$_2$ stored[12]. Here we report on an industrial-scale pilot project designed to find a carbon disposal solution for western Saudi Arabia. This arid region has large point-source CO$_2$ emitters, including petroleum refining and desalination facilities, but lacks saline aquifers and sedimentary traps[13–17]. We find that a CO$_2$ injection approach based on the recirculation of subsurface fluids can eliminate the need for external water. Our results demonstrate the feasibility of carbon mineral storage in regions in which access to water resources may be limited.

This project was designed to demonstrate a carbon disposal solution for the western region of Saudi Arabia. The demonstration site, located 24 km to the north-northeast of the Jizan Economic Complex and Refinery (Fig. 1), is underlain by the volcanic rocks of the Jizan Group. The Jizan Group comprises predominantly of 21–30-million-year-old bimodal volcanic and volcanoclastic rocks[15]. It was originally deposited in a continental rift valley during the early stages of the opening of the Red Sea basin[16]. The geologic history of the region has been summarized in detail[17]. The basalts of this region contrast with those previously used in Iceland for subsurface mineral carbon disposal, as the rocks in these previous pilot injections were far younger and/or less altered[6,11]. The Jizan Group is, at present, part of the eastern passive volcanic Red Sea margin. Such magma-rich volcanic margins contain dense fault[18] and fracture[19] networks. These provide secondary porosity and permeability for the injection of carbonated water. Estimates suggest that 4.2 Gt of carbon disposal through mineralization may be feasible in the basalts of the Jizan Group[20]. This carbon sink can also support the production of carbon-free fuels such as hydrogen and/or ammonia[21].

Five vertical wells, ranging from 100 to 1,000 m in depth, were drilled for the project (Fig. 1 and Extended Data Fig. 1). The top 30 m of the subsurface consists of post-rift conglomerate, sandstone and shale. These overlie, unconformably, a sequence of mildly altered but highly fractured basaltic lavas and dykes consisting of primary intermediate plagioclase and clinopyroxene and secondary chlorite, smectite, quartz, Fe–Ti oxides and minor calcite. The composition of these rocks is reported in Extended Data Tables 1 and 2. Two wells, DW-1 and DW-3, positioned 130 m apart, were designated as the production and injection wells, respectively (Fig. 1). These two wells were cased and cemented to 250 m below the surface (mbs) and completed as open holes below. Logging of these boreholes show that the basalts have substantial open horizontal fractures allowing subsurface fluid flow[19]. The other drilled wells showed no hydraulic connectivity to the DW-1 and DW-3 wells, so they were not sampled during this study. The production well was equipped with an electric submersible pump at 248 mbs. A 250-m-long, 2.0" stainless-steel pipe was installed inside the casing of the injection well to deliver downflowing carrier water supplied from the production well using a 2.0" nonmetallic (polyvinyl chloride, PVC) pipeline. A 0.5" stainless-steel pipe fitted at its bottom with a gas sparger was installed inside the 2.0" water supply pipe in the injection well. This 0.5" pipe carried pure CO$_2$, which was released into the carrier water stream as bubbles at a depth of 150 mbs. Another 3.0" 250-m-long stainless-steel pipe was installed inside the injection well for monitoring purposes. A submersible TV camera installed at the

[1]Institute of Earth Sciences, University of Iceland, Reykjavik, Iceland. [2]Physical Science and Engineering Division, King Abdullah University of Science and Technology, Thuwal, Saudi Arabia. [3]Environmental Protection Department, Saudi Aramco, Dhahran, Saudi Arabia. [4]Dipartimento di Scienze della Terra, Sapienza Università di Roma, Rome, Italy. [5]Warm Arctic ehf, Reykjavik, Iceland. ✉e-mail: eric.oelkers@gmail.com

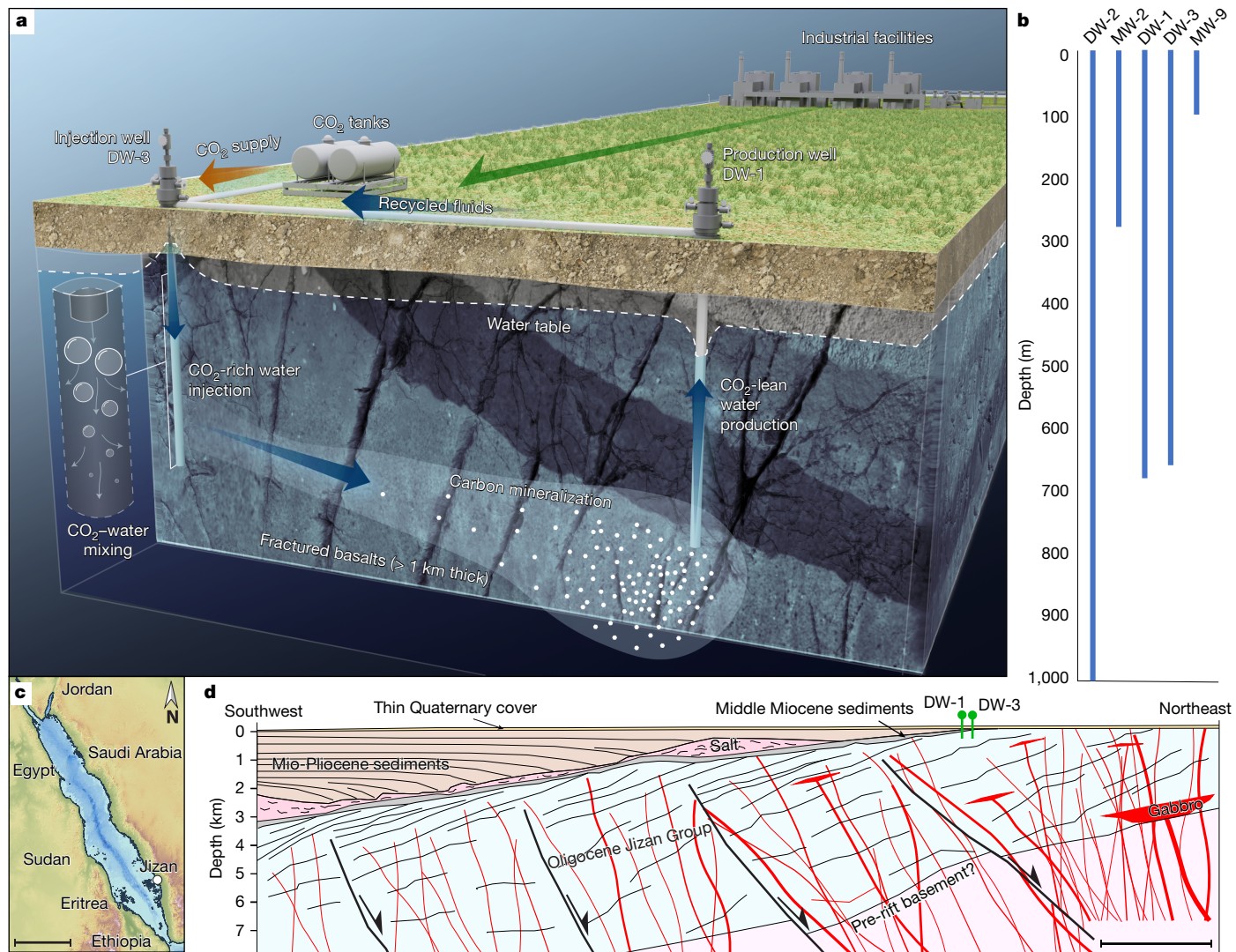

**Fig. 1 | Overview and layout of the Jizan CO₂ mineralization demonstration project. a**, Schematic representation depicting both the surface and the subsurface configurations of the pilot project featuring a water recirculation system (not to scale). Fluids collected from the DW-1 production well were reinjected into the DW-3 injection well, along with further CO₂ and chemical tracers, as detailed in the text. **b**, The depth intervals of the injection and production wells, as well as other wells drilled for the project. The relative location of these wells is shown in Extended Data Fig. 1, **c**, Map of the Red Sea showing the location of Jizan on the Saudi Arabian coast. Scale bar, 400 km. **d**, A schematic east–west geologic cross-section of the Jizan Group from our study area to the Red Sea. The Jizan Group consists of seaward-tilted half-grabens filled with layered volcanics, which show growth against antithetic northeast-dipping normal faults[18]. The Oligocene Jizan Group dips at an approximately 30° angle towards the west, covered by a Pliocene sedimentary cover at the study site. The sedimentary cover increases in thickness to the west. Scale bar, 4 km.

bottom of this pipe confirmed that the CO₂ released into the 2.0" pipe dissolved completely into the carrier water stream before reaching the open hole interval of the well[22].

## Carbon dioxide injection

Mineral carbon storage was achieved by the injection of water-dissolved CO₂ into the Jizan Group basalts. The dissolution of CO₂ in water assists this process in two ways[11,12]. First, water-charged CO₂ is denser than CO₂-free water of the same composition, leading to a non-buoyant injected fluid. This limits the risk of the CO₂ migrating towards the Earth's surface and its potential return to the atmosphere. Second, water-charged CO₂ is acidic and thereby accelerates greatly the dissolution of silicate minerals in the target host rock. The dissolution of silicate minerals in basalts provides both the divalent cations required for the formation of stable carbonate minerals such as calcite and the

alkalinity to the fluid, leading to the saturation and eventual precipitation of calcite and other carbonate minerals.

Water was pumped from the DW-1 production well and reinjected into the DW-3 injection well starting on 29 March 2023, at a flow rate of 2.6 kg s⁻¹; this flow rate is close to the injectivity limit of the well. Water was continuously circulated between the two wells until April 2024, except for 18 days in September 2023, when the subsurface pump in the production well malfunctioned and required replacement. Water circulation in this system was isolated from the atmosphere to avoid CO₂ leakage and to prevent the addition of O₂ to the subsurface. The addition of O₂ to the subsurface could oxidize iron and/or promote microbial activity that could block subsurface flow paths. Water circulation created a strong hydraulic gradient between the two wells. Before recirculation, the piezometric levels in both wells were approximately 40 mbs. After water circulation began, the water level in the production well decreased to 196 mbs and that of the injection well increased to

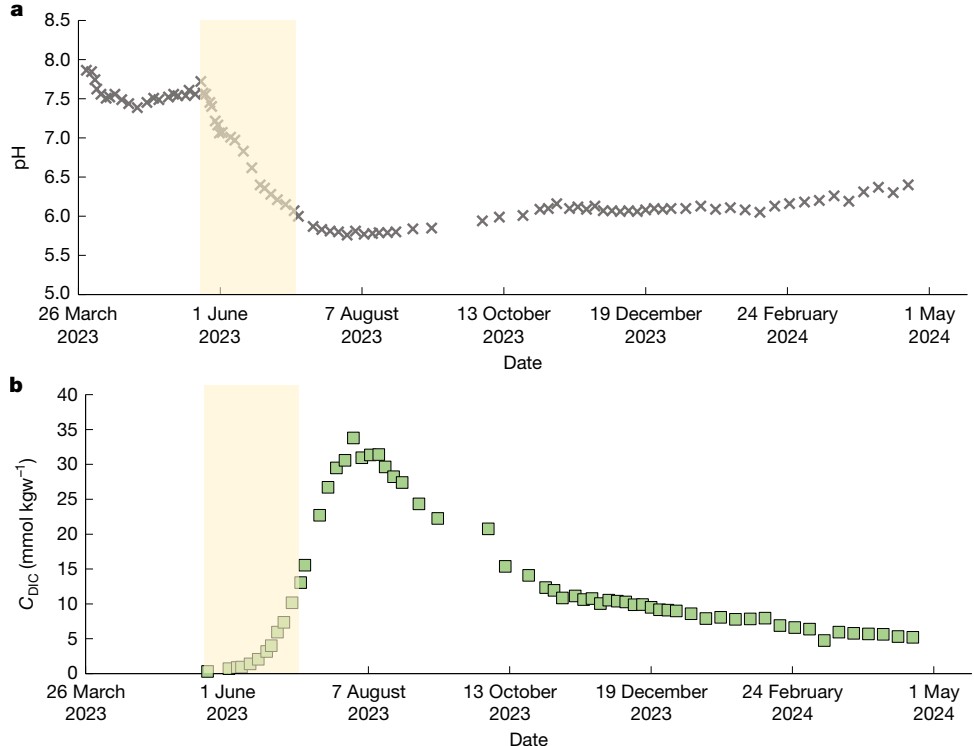

**Fig. 2 | Temporal evolution of the in situ pH and DIC concentration before, during and after the injection of 131 tons of water-dissolved $CO_2$ in the injection well. a**, In situ pH at 45 °C. **b**, DIC concentration. The yellow shaded areas show the period during which $CO_2$ was injected into the system. The uncertainties of the data, equal to two times the standard deviation of repeated analyses, are smaller than the symbol size.

17 mbs. The use of a recirculation system, which makes use of water sourced from the injection zone, provides substantial advantages. First, it removes the need to deliver external water to the site. Second, it reduces the risk of fluid pressure build-up in the subsurface. Third, because the injected water has the same composition as that of the subsurface reservoir, it reduces the risk of compatibility issues such as scaling or loss of reservoir permeability. Fourth, mixing reservoir water with $CO_2$ produces an injection fluid of higher density than the original reservoir water, hence minimizing the potential for $CO_2$ exsolution.

The characterization of the subsurface hydraulic system involved injecting a 2.3-kg slug of sodium fluorescein (NaF) tracer into the circulating water system on 5 April 2023. Our analysis indicates that fluid flow occurs through two distinct pathways: a primary high-permeable natural fracture network resulting in a fast breakthrough at the production well and a secondary flow system, comprising most of the pore volume through a matrix, including smaller interconnected and less permeable fractures. The analysis estimated a total effective pore volume of 24,000–43,000 $m^3$, with 10% corresponding to fast-flow regions with a residence time of 50–65 days and 90% corresponding to the low-permeability matrix with a residence time of 255–445 days. The flow primarily occurred along the main faults, whereas the matrix served as a secondary flow conduit with a longer residence time. This is consistent with the natural fracture and fault systems identified from a detailed analysis of wellbore image logs acquired in the wells as part of the data acquisition and logging campaign[19].

The $CO_2$ injection system underwent testing from 16 to 25 May, during which limited amounts of $CO_2$ were intermittently injected into the subsurface. Continuous $CO_2$ injection started on 31 May 2023, by releasing pure $CO_2$ at a pressure of 12–14 bar into the stream of carrier water flowing down the injection well. Both electronic and mechanical flow meters measured the water circulation rate from production well DW-1, whereas a gas flow meter provided the mass flow rate of $CO_2$. A carbon dioxide to water mass ratio of 1:65 was chosen so that $CO_2$ will be slightly below its aqueous solubility at the 45 ± 0.5 °C temperature and the pressure of the shallowest target injection depth of 350 m. The inert chemical tracer $SF_6$ was co-injected into the subsurface at a constant $SF_6$-to-$CO_2$ mass ratio of $1.05 \times 10^{-6}$:1 to help assess the fate of the injected $CO_2$. In total, 137 g of $SF_6$ and 131 tons of $CO_2$ were injected into the subsurface by 7 July 2023, when the $CO_2$ injection was stopped, whereas water circulation continued for approximately another ten months. Fluid samples were collected daily at the production well from 29 March until 31 December 2023 and then every 4 days thereafter until 21 April 2024. The samples were analysed for in situ pH, alkalinity and redox state, as well as elemental and chemical tracer concentrations.

The temporal evolution of the pH and the dissolved inorganic carbon (DIC) concentration of the production well fluids are shown in Fig. 2. The pH of the formation fluids stabilized at 7.5 before the start of $CO_2$ injection. The DIC concentration of the production well fluid was 0.31 mmol $kgw^{-1}$ before the $CO_2$ injection. This concentration increased concurrently with a decrease in pH as the injected carbon reached the production well. The DIC concentration maximized and the pH minimized on 7 August 2023. Subsequently, the production well fluid pH increased whereas the DIC concentration steadily decreased. The maximum recorded DIC concentration of the production well fluids was 33.8 mmol $kgw^{-1}$. This is approximately 90% less than the DIC of the injected $CO_2$ at 350 mmol $kgw^{-1}$. This notable DIC concentration reduction highlights the capacity of the system to effectively reduce DIC levels through fluid-circulation-related dilution and mineral reactions.

## Quantifying the fate of the injected $CO_2$

The production well fluids were supersaturated with respect to calcite and undersaturated with respect to plagioclase and pyroxene before the dissolved $CO_2$ injection. This probably reflects the flushing of the system, which mixed some shallower lower-temperature fluids into the production well. After the injection, as the $CO_2$ plume arrived at

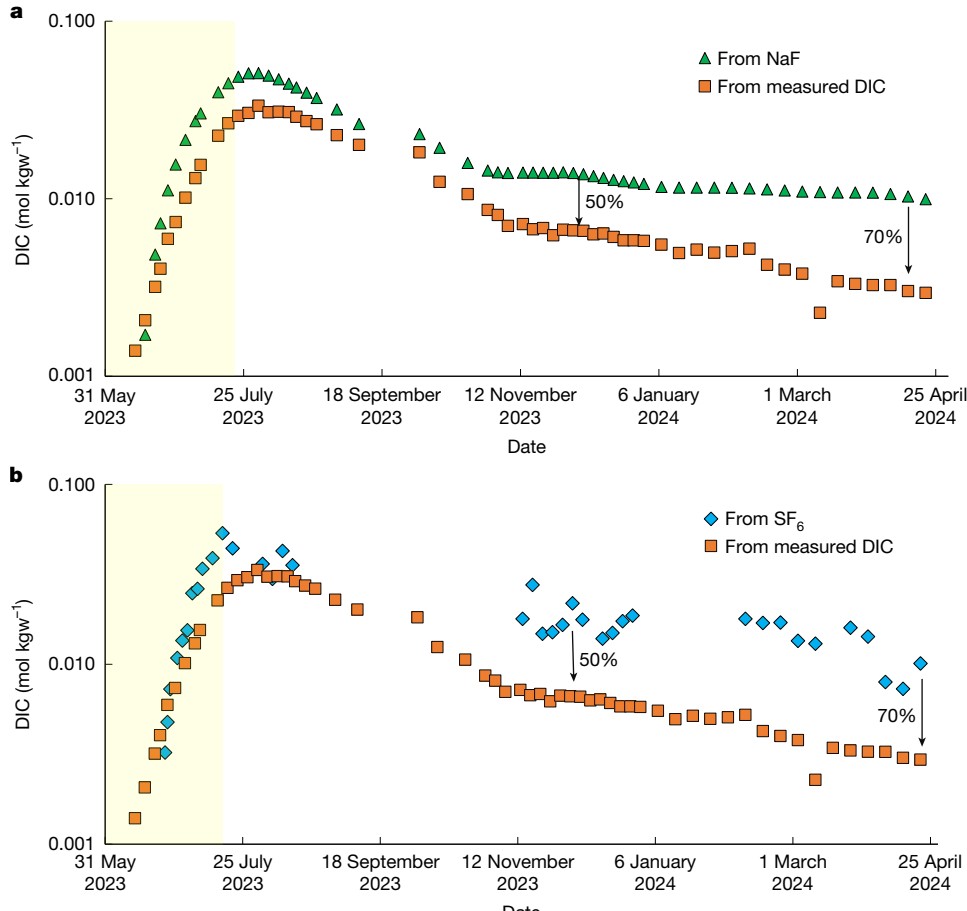

**Fig. 3 | Estimates of carbon mineralization fraction over time. a**, Comparison of measured DIC to the expected DIC baseline, derived using the NaF tracer. This shows a gradual reduction to 50 ± 5% by November 2023 and 70 ± 5% by April 2024, suggesting that approximately 70% of the carbon was mineralized after 330 days of $CO_2$ injection and recirculation. **b**, Comparison of measured DIC to the expected baseline using the $SF_6$ tracer. This indicates a reduction in DIC of approximately 50 ± 5% by November 2023 and 70 ± 5% by April 2024, aligning with the results obtained from the NaF tracer. The yellow shaded areas show the period during which $CO_2$ and $SF_6$ were injected into the system. The analytical uncertainties in the data are approximately equal to the symbol size.

the production well, the concentrations of dissolved Si, Mg and Ca increased by factors of 2, 3 and 1.25, respectively (Supplementary Table 1 and Extended Data Fig. 2). This is consistent with the dissolution of silicate and/or carbonate minerals in the subsurface basalt storage formation. The dissolution of silicate minerals is also supported by the saturation states of the primary plagioclase and pyroxene in the sampled production well fluids (Supplementary Table 2 and Extended Data Fig. 3). These minerals are strongly undersaturated in the production well fluids during and after the $CO_2$ injection. The dissolution of calcite near the injection well is likely, as the injected $CO_2$-charged water is strongly undersaturated with respect to calcite and because calcite dissolves orders of magnitude faster than the silicate minerals present in the subsurface[23]. The production well fluids had much lower DIC concentrations than those of injected fluids and the production well fluids are saturated with respect to calcite. This indicates that further calcite precipitated beyond what initially dissolved as the injected fluids travelled in the subsurface.

Following the peak DIC concentration on 7 August, 68 days after the beginning of the $CO_2$ injection, the concentrations of DIC, Ca, Mg and Si in the reservoir fluid gradually declined. Although some of this could be the result of fluid dilution, these concentrations decrease towards their background values faster than the injected conservative tracers. This suggests that precipitation reactions occurred in the subsurface. The likely precipitation of carbonate minerals at this time is suggested by their saturation states in the production well fluids

(Extended Data Fig. 3). The common carbonate mineral, calcite, was close to saturation throughout the study period. Despite the presence of $Mg^{2+}$ in the produced water, calcite formation rather than aragonite formation was favoured, as aragonite is calculated to be undersaturated in nearly all of the collected fluids. The carbonate mineral ankerite $(Ca(Fe,Mg)(CO_3)_2)$ was strongly supersaturated and siderite $(FeCO_3)$, was mildly undersaturated after 4 July, coinciding with an increase in dissolved Fe concentration. The precipitation of calcite, ankerite and siderite is supported by the simultaneous decrease of Ca, Fe and DIC concentrations. The decrease in Si, Mg and Fe could also result from clay mineral precipitation. The smectite clays montmorillonite and beidellite were strongly supersaturated in the production well fluids (Extended Data Fig. 3). By contrast, all zeolite minerals and magnesite were undersaturated in the production well fluids, so they are unlikely to form. The conclusion that subsurface carbon fixation is dominated by mineralization rather than by biotic carbon production, as suggested by some researchers[24], is supported by the invariant injectivity of the Jizan pilot over time; subsurface biotic carbon production has been observed to decrease injectivity substantially at the CarbFix1 site[12]. Furthermore, an inspection of a damaged submersible pump recovered from the production well on 10 September 2023 showed no evidence of biological material deposition.

Further evidence supporting the mineralization of the injected $CO_2$ is based on the analysis of the solids collected from the damaged submersible pump recovered from the production well.

The inside of the pump was filled with grains of the original subsurface rock formation cemented by up to 14 mass% of calcite and up to 4% siderite and 3% ankerite (Extended Data Table 3). Carbonate scaling was also observed on the surface of the recovered pump. The $\delta^{13}C$ and $\delta^{18}O$ compositions of the carbonate cements recovered from the pump ranged from −7 to −28‰ and from −20 to −11‰, respectively (Extended Data Fig. 4). The lowest $\delta^{13}C$ values and the highest $\delta^{18}O$ values correspond to samples containing the highest carbonate concentration. The $\delta^{13}C$ composition of the DIC in the formation waters before the injection of $CO_2$ was −12.9‰, whereas the $\delta^{13}C$ composition of injected $CO_2$ was −37.1‰. The $\delta^{18}O$ of the circulating water was −3.3‰. Estimates based on the equilibrium carbon and oxygen stable isotope fractionation factors in the $H_2O$–$CO_2$–calcite system[25,26] indicate that the adhering solids are cemented by $CO_2$ injected during the pilot test. The variability of the $\delta^{13}C$ and $\delta^{18}O$ values of the samples indicates the presence of some detrital carbonate component in the samples.

The mass of $CO_2$ mineralized during this study was characterized using two independent methods (Extended Data Figs. 5 and 6). The first is through a comparison of the DIC concentrations with those of the injected NaF. This inert tracer has been used extensively to trace groundwater and has been found to be conservative in subsurface environments[27,28]. The initial 2.3-kg NaF slug was recirculated continuously during and following the $CO_2$ injection. The second method compares the corresponding production well $SF_6$ and DIC concentrations. The latter approach was previously used by CarbFix to quantify the rate of subsurface mineral carbonation during their $CO_2$ injection[11]. These two approaches provide a baseline of DIC concentrations expected in the absence of chemical reactions. The ratio of the DIC measured in the production well fluids to the baseline calculated using these chemical tracers provides an estimate of the percentage of $CO_2$ mineralized from the fluid phase over time (Fig. 3). The analytical and computational details for each approach are provided in Methods.

Despite some scatter, both independent tracers are consistent with the mineralization of the injected $CO_2$ over time. Mineralization was first apparent after 1 August, approximately one month after the end of the water-dissolved $CO_2$ injection. About $70 \pm 5\%$ of the injected $CO_2$ is estimated to have been mineralized by 21 April 2024 according to the NaF and $SF_6$ tracers. This value is based on the change in the proportion of the tracer concentrations compared with the measured DIC. Although much of this carbonate may be calcite, the presence of ankerite and siderite in our system (Supplementary Table 3) suggests that subsurface carbonation of basalts at a pH of about 6 could produce carbonate minerals containing Fe and Mg. Note that Mg is commonly found in ankerite and siderite precipitated in basaltic terrains[29]. The uptake of Fe and Mg by carbonate minerals can slow the formation of divalent metal clay minerals, resulting in a more efficient use of liberated divalent cations and, potentially, subsurface pore space.

## Discussion

The results of this study indicate that $70 \pm 5\%$ of the $CO_2$ injected into the subsurface Jizan formation mineralized within ten months. This was achieved by the recirculation of $CO_2$-charged water between two closely spaced wells drilled into this 21–30-million-year-old basalt. The upscaling of this technology in the Jizan region of Saudi Arabia, in which subsurface basaltic formations are present, could sequester large amounts of $CO_2$ (see Methods for storage capacity estimates). The application of a scalable multi-well-based water recirculation infrastructure in the region provides a robust approach to expanding $CO_2$ mineral storage operations without the need for extensive external water. This approach thereby provides the means to adapt the process of subsurface $CO_2$ storage through mineral storage in basaltic terrains in regions in which water may be scarce, such as the Middle East, in which much of global oil production is located.

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

# Methods

## Chemical composition of the target subsurface basalts

Cuttings collected from the injection and production wells were characterized by X-ray fluorescence (XRF) to estimate their chemical compositions. The XRF measurements were performed at the laboratories of Isotope Tracer Technologies Europe (IT$^2$E) in Milan, Italy and at King Abdullah University of Science and Technology (KAUST) in Thuwal, Saudi Arabia.

At the IT$^2$E laboratory, the samples were pulverized with an agate mortar and pestle and their volatile contents were determined by the loss on ignition method. The powders were then dried in an oven at 110 °C overnight, heated in a muffle oven to 1,000 °C, mixed with powdered boric acid and compressed into pellets. The pellets were loaded onto an ARL automatic ADVANT XP spectrometer equipped with a Rh front-window X-ray tube. Analyses were performed using an applied power of 3.0 kW. The count times on identified peaks were 10 s for major elements and 40 s for trace elements. Matrix correction and interelement effects were accounted for using the method of Lachance and Traill[30]. Analytical uncertainties were determined by analysis of international standards with known compositions. For the major elements, the uncertainties in Si, Ti, Ca and K are less than 3%, whereas for Al, Mn, Mg, Na and P, the uncertainties are less than 7%.

The samples analysed at the KAUST laboratories were first ground to a powder. Approximately 1 g of each sample was mixed with 9 g of XRF Scientific X-ray flux powder composed of 66% lithium tetraborate and 34% lithium metaborate. This mixture was melted at 1,050 °C in an Eagon 2 fusion instrument and poured to form a homogenous pellet that was analysed on a Bruker S8 TIGER machine. The detection limit was <150 ppm for all elements and the uncertainty was <1% for the major elements.

The mineral phases present in the ground well cutting samples were characterized by X-ray diffraction (XRD) analyses conducted at the Exploration Core Labs Department (ECLD) of Saudi Aramco and at IT$^2$E. The XRD analyses at ECLD were run on powders ground with an agate mortar and pestle using a Rigaku Ultima IV powder X-ray diffractometer with CuKα radiation (40 kV, 40 mA) over the 3°–70° (2$\theta$) interval, with a step size of 0.02° increment and a scan speed of 1° s$^{-1}$. We interpreted the XRD patterns using X'pert HighScore software using specific crystallographic information files for Rietveld refinement and conducted cluster analysis using JADE Pro and its toolkit.

The XRD analyses at IT$^2$E were performed on samples dried at 60 °C and then ground in an agate mortar and pestle. Each sample was then loaded on a onto a polymethylmethacrylate (PMMA) sample holder and placed in a Bruker D8 ADVANCE DaVinci automatic powder diffractometer, equipped with a LYNXEYE detector set to discriminate CuKα$_{1,2}$ radiation. The interpretation of the diffractogram for phase identification was carried out by comparison with crystalline phases of the PDF-2 International Centre for Diffraction Data (ICDD) and Crystallography Open Database (COD) databases. A preliminary semi-quantitative estimate of weight fractions was carried out by using the normalized reference intensity ratio method (also known as the Chung method[31]). This method uses scaling factors assigned according to the heights of the characteristic peaks of the different phases. No internal standard was added, so the presence of any amorphous phase was not checked. Quantitative phase analysis of each sample was achieved by the Rietveld profile fitting method as implemented in the Bruker TOPAS V.5 program. This is based on the fundamental parameters approach[32]. The crystal structure models of crystalline phases considered in the XRD profile fitting for andesine plagioclase, augite pyroxene, clinochlore chlorite, montmorillonite, quartz, laumontite zeolite, calcite and richterite amphibole were taken from the Inorganic Crystal Structure Database (ICSD) Release 2021-2 (ref. 33) (Supplementary Table 4). Unit cell parameters, scale factors and crystal

sizes were allowed to vary for all phases. Atomic coordinates and atomic displacement parameters were fixed while site occupancy factors of octahedral cations and extra-framework species were adjusted, with restraints, in richterite and laumontite, respectively, to account for the crystal-chemical variations of these phases compared with the model. Rietveld profile fitting allowed testing of the presence of mineral phases preliminarily identified and semi-quantified by the reference intensity ratio method[34]. The resulting chemical and mineralogical compositions of the subsurface drill cuttings are provided in Extended Data Tables 1 and 2.

## Chemical analysis of production well fluid samples

Fluid samples were regularly collected from a dedicated outlet port located at the production well. Fluid temperature, pH at the reservoir temperature of 45 ± 0.5 °C and total dissolved solids were measured directly at the fluid sampling port using a Myron L Ultrameter II 6PFC multimeter. The pH electrode was regularly calibrated using Mettler Toledo pH 4.01, 7.00 and 10.01 standard buffer solutions. The uncertainty of the pH measurements was ±0.02 based on replicate analyses of the standard buffer solutions. Measured on-site pH values are reported in Fig. 2. Samples for alkalinity measurement were collected in cleaned 500-ml polyethylene terephthalate (PET) bottles. Further samples were immediately acidified, after filtering using a 0.22-μm syringe filter, by adding 2–3 drops, or approximately 0.01 ml of double-distilled nitric acid, containing 67–69% by weight $HNO_3$, to 50-ml production well samples in cleaned PET bottles. The PET bottles originally contained drinking water but were rinsed several times first with the production well water before sampling. Both sets of samples were stored in an insulated cooler until transported for chemical analysis. The alkalinity of the first fluid sample was measured by acid titration using the Gran function plot method[35]. The measured alkalinity was then used to calculate the DIC concentration using PHREEQC[36] together with the measured pH and fluid compositions of the major elements at the 45.5 °C subsurface temperature. PHREEQC is a geochemical modelling code designed to perform a variety of aqueous geochemical calculations, including calculations of saturation indices. The second fluid sample was analysed for major cation and Si concentrations by inductively coupled plasma optical emission (ICP-OES) spectroscopy using an Agilent 5110 ICP-OES at KAUST. This spectrometer analysed the compositions of Fe, K, Mg, Si, Ca and Na using wavelengths of 239.563, 769.879, 279.800, 251.611, 315.887 and 568.821 nm, respectively. This instrument was calibrated using Sigma-Aldrich ICP standard solutions of 0.1, 1, 10 and 100 ppm concentration. The limits of detection for Ca and Si were 0.01 and 0.02 ppm, respectively. The analytical uncertainty was <0.1 ppm for all measured elements. To assess the potential contamination from the use of the PET bottles used in fluid sampling, three blanks were prepared using distilled demineralized water. These blanks were prepared and analysed identically to those used for the production well sampling. In each case, the concentration of all measured elements in the prepared blanks was below the respective analytical detection limits. Measured concentrations over time of all sampled fluids are illustrated in Extended Data Fig. 2 and tabulated in Supplementary Table 1.

On the basis of the ICP-OES, pH and alkalinity measurements, the saturation state of the sampled fluids with respect to selected minerals was calculated using PHREEQC[36] with its Kinec_v3 database[23]. The Kinec_v3 database is the most recently updated database for use with PHREEQC. As the concentration of Al was below the analytical detection limit of our measurements, the concentration of this element was set to be in equilibrium with diaspore in the calculations. This choice was made as diaspore is readily observed to precipitate during experimental studies of basalt dissolution and that chlorite and zeolite minerals, such as clinochlore and laumontite, are more soluble than diaspore at our field conditions. The resulting saturation indices are provided in Supplementary Tables 2 and 3 and Extended Data Fig. 3.

## Carbon isotope measurements

Carbon isotope compositions in this manuscript are presented in the delta notation given by:

$$\delta^{13}C\ (‰) = 10^3 \times \left( \frac{^{13/12}C_{Sample}}{^{13/12}C_{V\text{-}PDB}} - 1 \right),$$

in which $^{13/12}C$ refers to the indicated molar $^{13}C$ to $^{12}C$ isotope ratio, $\delta^{13}C$ provides the normalized value of this ratio and the subscripts Sample and V-PDB represent the sample of interest and the V-PDB international standard, respectively.

The oxygen isotope values of the solid carbonates are also reported according to the V-PDB standard as

$$\delta^{18}O\ (‰) = 10^3 \times \left( \frac{^{18/16}O_{Sample}}{^{18/16}O_{V\text{-}PDB}} - 1 \right)$$

Carbon and oxygen stable isotope compositions of solid carbonate samples and carbon isotope compositions of the $CO_2$-injected gas were determined at IT$^2$E. The stable isotope analyses were conducted using a Finnigan MAT Delta$^{plus}$ isotope mass spectrometer using the NBS-18, NBS-19, IT2-20 and IT2-21 standards. The standard deviation associated with the measurements is ±0.3‰ for $\delta^{13}C$ and ±0.2‰ for $\delta^{18}O$. The $\delta^{13}C$ and $\delta^{18}O$ compositions of the carbonates recovered from the disabled production well pump are reported in Extended Data Table 3.

## Identification of solids collected from the damaged submersible pump

Fourteen solid samples were removed from the damaged submersible pump and one sample of the dust collected from this pump while drying was analysed for their mineralogical and isotopic composition. The results are provided in Extended Data Table 3. All XRD patterns collected for this purpose are provided in Supplementary Fig. 1.

The $\delta^{13}C$ composition of the injected $CO_2$ gas was −37.1‰ and that of the formation fluids before the injection of the $CO_2$ was −12.9‰. The $\delta^{18}O$ of the circulating water was −3.3‰. Taking account of the equilibrium fractionation factors of these systems[25,26], the $\delta^{13}C$ and $\delta^{18}O$ values of calcite in isotopic equilibrium with the injected $CO_2$ and the local formation fluid at 46 °C would be −29.0 and −10.0‰, respectively. These values are close to the $\delta^{13}C$ and $\delta^{18}O$ values of the recovered disabled pump samples having the highest amounts of carbonate as determined by XRD (Extended Data Fig. 4). This indicates that the cements are fresh carbonate that precipitated from the injected $CO_2$ and local groundwater. The variability of the $\delta^{13}C$ and $\delta^{18}O$ values of the rest of the pump samples indicates that, as well as fresh carbonate, the samples also contain a detrital carbonate component (Extended Data Fig. 4). The $\delta^{13}C$ and $\delta^{18}O$ compositions of this detrital component fall within the ranges of the $\delta^{13}C$ and $\delta^{18}O$ compositions of carbonates extracted from quartz-carbonate veins associated with the hydrothermal alteration of the basalts, which vary from −3.2 to −6.7‰ and from −19.7 to −21.8‰, respectively (Extended Data Fig. 4).

We attribute the precipitation of carbonate cements on the submersible pump to the supersaturation of carbonate minerals through subsurface mineral reactions rather than degassing of the fluids in the subsurface. This is because the partial pressure of $CO_2$ in the production well fluids did not exceed 1 bar, whereas the pressure in the submersible pump was not less than 5 bar. Thus, there was no driving force for the degassing of $CO_2$ within the pump during our study. Furthermore, no $CO_2$ gas was observed in the recirculated fluid at the injection well, which included a gas trap and a $CO_2$ concentration sensor at the wellhead.

## Calculation of the expected DIC concentration in the absence of chemical reaction

The percentages of injected $CO_2$ fixed into solid phases by chemical reactions in the subsurface during this study are determined by comparing DIC concentrations measured in the production well fluids to estimates of the DIC concentration expected in the absence of subsurface reactions. Two independent approaches were used to calculate the expected DIC concentrations in the absence of subsurface reactions. These two approaches are described below.

**From NaF.** NaF was added to the recirculating fluid as a single slug of 2.3 kg on 4 April 2023 to characterize the subsurface flow paths at the pilot test site. The concentration of NaF was then continuously monitored in the production well fluids. Samples for NaF determination were collected in clean PET bottles and stored in the dark until analysis. The concentrations of NaF were measured at KAUST by first adjusting all samples to a pH > 8.7. This was achieved by adding a pH 9 $NH_4Cl$–$NH_4OH$ buffer solution to the original samples at a 1:1 ratio. The diluted samples were then analysed using a Cary Eclipse Fluorescence Spectrometer at a 512-nm wavelength. The spectrometer used a spectral bandwidth of 5 nm and an excitation of 5 nm. This system has a detection limit of 0.1 ppb and an uncertainty of approximately 1%, as determined by replicate analyses. The measured NaF concentrations at the production well over time are shown in Extended Data Fig. 5a. The NaF breakthrough was detected in the production well fluids 8 days after its addition to the injection well. The NaF concentration then increased to a maximum of 44 ppb on 6 May 2023. This concentration then decreased as the fluids were continuously recycled into the subsurface system. The DIC measured at the production well (Extended Data Fig. 5b) showed a trend similar to that of the NaF tracer, peaking at approximately 33.8 mmol kgw$^{-1}$. This peak value is 90% lower than the DIC injected into the injection well, which was around 350 mmol kgw$^{-1}$.

To analyse the NaF tracer and DIC production profiles, the first-moment analysis technique was applied. This method adjusts for the recycled chemicals within the production curves by incorporating the age distribution, also known as the residence time distribution, $E(t)$ (refs. 37–41), defined as

$$E(t) = \frac{Q_p C_m(t)}{m_{inj}}$$

in which $Q_p$ represents the fluid production rate (m$^3$ day$^{-1}$), $C_m(t)$ denotes the chemical concentration measured at the production well (kg m$^{-3}$) and $m_{inj}$ designates the total mass of the chemical injected (kg). The unit of $E(t)$ is inverse time (day$^{-1}$). The first moment, $\tau$, is used to calculate the mean residence time between the injection and production wells, defined by

$$\tau = \int_0^\infty t E(t) \, dt$$

Because the chemicals are reinjected, the measured concentration history reflects a combined effect of the initial injection and the continuing recycling of the produced injectate. Moment analysis, which addresses the response to a one-pass slug injection, requires the removal of the recycling effect to accurately determine chemical residence time and swept pore volumes. The convolution integral can be applied to isolate the response of the initial tracer injection, resulting in a single-pass tracer return profile[37]. The adjusted concentration, $C_{adj}(t)$, thus represents the concentration at which the chemical is removed from the production fluids before being reinjected, such that

$$C_{adj}(t) = C_m(t) - \int_0^t C_m(\tau) C_m(t - \tau) \, d\tau$$

To account for the effects of fluid recycling, the deconvolution operator was applied to both the NaF tracer and the DIC concentration profiles. This step was necessary to isolate the first-pass breakthrough of the injected fluid from subsequent recirculation. The resulting adjusted concentrations of the NaF tracer and DIC concentration are shown in Extended Data Fig. 5a,b, respectively. The difference between the measured concentrations and these adjusted profiles is attributable to the recirculation of the chemicals.

Because NaF was injected as a single pulse, whereas $CO_2$ was injected continuously, an extra adjustment was required to ensure that the two chemicals share the same injection profile for a direct comparison. To achieve this, the continuous-response NaF concentration, denoted as $C_{Cont,NaF}(t)$, was calculated by convolving the pulse injection profile of NaF with the continuous $CO_2$ injection function, $Inj_{CO2}(t)$. Therefore, the convolution operator was applied to the pulse NaF injection profile, $C_{Pulse,NaF}(t)$, to ensure that the response of NaF mimics a continuous injection scenario, such that

$$C_{Cont,NaF}(t) = \int_0^t C_{Pulse,NaF}(\tau) Inj_{CO2}(t - \tau) d\tau$$

The effective pore volume, PV, swept by these chemicals can be estimated from

$$PV = \frac{Q_P^2}{m_{inj}} \int_0^\infty t C_{adj}(t) dt$$

Given that NaF is a conservative and non-reactive tracer, an estimate of the DIC concentration was made by assuming that carbon behaves similarly to NaF in the absence of subsurface reactions. This assumption requires that diffusion of $CO_2$ and the tracer if it diffuses into the rock matrix do so at the same rate. This would be likely, as they are both dissolved aqueous phases. The residence time distribution for NaF was then convolved to match the injection profile of $CO_2$. The calculated DIC concentration, $DIC_{cal,NaF}$ (mol kgw$^{-1}$), in the production well fluids in the absence of subsurface reactions, estimated from the convolved NaF profile, $E_{NaF}(t)$, is given by

$$DIC_{cal,NaF}(t) = \frac{M_{CO2}}{Q_p} E_{NaF}(t)$$

in which $M_{CO2}$ is the total mass in mol of the injected $CO_2$.

The expected DIC concentrations in production well fluids, assuming no subsurface reactions, are compared with the measured DIC concentrations in Fig. 3a. This comparison is used to estimate the percentage of $CO_2$ mineralized. The data indicate progressive mineralization of the $CO_2$ up to 70 ± 5% within eight months after stopping $CO_2$ injection.

Several studies suggest that NaF can sorb onto negatively charged silicate mineral surfaces at neutral to acid pH, leading to the non-conservative behaviour of this tracer[42,43]. Notably, one study[43] found that the concentration of NaF decreased substantially in formation waters when co-injected together with $CO_2$, $NO_2$, $SO_4$ and $O_2$ into the subsurface. This study concluded that NaF was not suitable as a tracer to quantify the fate of $CO_2$ injected into geologic formations and ascribed this decrease to the sorption of the tracer to mineral surfaces. Two lines of evidence indicate that such an effect did not occur in our study, First, most of the NaF injected during our study is accounted for in the recovered production well fluids. Second, the fraction of carbon mineralized determined from NaF and $SF_6$ are identical, suggesting that any effect of NaF sorption on subsurface mineral storage in our study region is negligible (Fig. 3). In this regard, it should be noted that, if some NaF had selectively sorbed onto subsurface minerals in our study, this process would have lowered the concentration of recovered NaF and thus the estimated DIC concentration in Fig. 3a. As a consequence, if selective NaF sorption had occurred, the percent carbon

mineralized in this study would have been underestimated and the true percentage of $CO_2$ mineralized would have been higher than reported in the main text. The difference in NaF behaviour in the present study compared with the previous study[43] could potentially be attributed to the co-injection of oxic gases with the NaF tracer. NaF is commonly known to oxidize to a non-fluorescent product in the presence of a variety of oxidizing agents[44].

**From $SF_6$.** Following the approach of Matter et al.[11], $SF_6$ was used as an inert tracer to estimate the percentage of the $CO_2$ mineralized over time. To this end, $SF_6$ was co-injected with $CO_2$ at a constant $SF_6$ to $CO_2$ mass ratio of $1.05 \times 10^{-6}$:1. Both the $CO_2$ and the $SF_6$ injection were stopped on 7 July 2023. The $SF_6$ was originally stored in gas cylinders. The gas was added to the recirculating water stream using a mechanical flow regulator connected to an AALBORG electronic mass gas flow meter recalibrated for $SF_6$. Fluid samples for $SF_6$ analysis were collected in pre-cleaned 1.0-l glass bottles provided by Spurenstofflabor. The sample bottles were completely submerged in a 25-l plastic container fed with water from the sampling port of the production well for sample collection. Concentrations of $SF_6$ were measured at Spurenstofflabor. Sample preparation included the extraction of a 40-ml aliquot injected into glass bottles filled with $N_2$ gas. After equilibration by shaking for 30 min, 10 ml of headspace gas was separated for further dilution with $N_2$ gas. Aliquots were analysed by gas chromatography using an electron capture detector. The detection limit is below 0.1 fmol l$^{-6}$ $SF_6$. The reproducibility is <2% for air samples and <10% for water samples. The reliability of the measurements was determined by analysing two random duplicate samples (that is, DW-3: 6 July and DW-1: 9 August and 17 and 24 March) with and without the final dilution step. The differences between the results were always less than 10%.

The measured $SF_6$ concentrations are provided in Fig. 3b. The $SF_6$ concentration in the production well waters first increased, maximizing during 9–12 July, or shortly after the $CO_2$ and $SF_6$ injection was stopped. A slow decline in the concentration is observed after this time. This can be attributed to a continuous dilution of this tracer over time. Dilution of the fluid is favoured owing to the lowering of the water table at the production well. The water table at the production well decreased by approximately 150 m owing to pumping. This lower water table height drives groundwater from surrounding subsurface to the production well as well as the injection well, diluting the concentrations of both $SF_6$ and NaF.

The measured $SF_6$ concentrations are compared with the corresponding DIC concentrations in Fig. 3b. Similar to the analysis for NaF tracer, the $SF_6$ tracer is non-reactive, therefore an estimation of DIC concentration was made assuming that carbon behaves similarly to $SF_6$ in the absence of subsurface reactions. The calculated concentration, $DIC_{cal,SF6}$ (mol kgw$^{-1}$), in the production well fluids, in the absence of subsurface reactions, is estimated from the adjusted residence time distribution of $SF_6$, $E_{SF6}(t)$, such that

$$DIC_{cal,SF6}(t) = \frac{M_{CO2}}{Q_p} E_{SF6}(t)$$

The calculated $DIC_{cal,SF6}$ concentration matched closely with that of the observed DIC until September 2023. Afterwards, the observed DIC concentration decreased faster with time than that of $SF_6$. As the $SF_6$ is non-reactive, the difference between the two curves in Fig. 3b can be attributed to the precipitation of the injected carbon in the subsurface, indicating about 70 ± 5% mineralization by April 2024, as shown in Fig. 3b.

### Reservoir characterization from the NaF tracer

The NaF tracer concentrations recovered from the production well revealed a complex distribution of fractures and matrix systems within the reservoir. This concentration profile suggests the presence of at

least two distinct flow systems in the reservoir with distinct permeabilities. We conducted a detailed analysis to compare the observed tracer profiles against simulations. Initial attempts to match the tracer data using a single-channel model were unsuccessful[41], as shown in Extended Data Fig. 6a. However, when a dual-channel model was applied, the tracer profile could be accurately matched (Extended Data Fig. 6b). This dual-porosity scenario implies that fluid flow within the reservoir occurred through two main pathways: (1) highly permeable conduits such as faults, which led to a rapid tracer breakthrough at the production well, and (2) through a more extensive network in the matrix composed of smaller, interconnected and less permeable fractures.

To quantify the effective swept pore volume, the method of moments was used[38]. This approach uses moment analysis, requiring a full production profile of the tracer over time. The extended tracer concentration profile at the production well was estimated using an exponential decline model, which, when plotted on a semi-log scale, results in a straight line, as depicted in Extended Data Fig. 6c. The outcome of this analysis, coupled with an uncertainty analysis, revealed that the total effective pore volume ranged between 24,000 and 43,000 $m^3$. Of this, approximately 10% is attributed to the fast-flow regions with a mean residence time of 50–65 days, whereas the remaining 90% is associated with the low-permeability medium, with residence times spanning 255–445 days.

In total, 1.55 kg of NaF was recovered from the production well fluids over the course of this study, up until 21 April 2024. After accounting for that reinjected through the circulation of the production well fluids, the adjusted NaF recovery was 1.06 kg. This compares with a total of 2.30 kg of NaF originally injected into the system. Thus, 46% of the originally injected NaF was recovered before 21 April 2024. It seems likely that the unrecovered tracer is primarily related to fracture–matrix diffusion and not sorption. The tracer initially diffuses from fractures into the low-permeability basalt matrix and then returns slowly from matrix to fractures over multi-month timescales. Two independent observations support this interpretation. First, the adjusted cumulative mass of NaF continues to increase for months after the breakthrough peak, indicating continuing release of stored tracer. Second, a semi-log linear tail of the recovery of this tracer is observed (Extended Data Fig. 6c). Such a single-timescale exponential tail is the classic signature of mobile (fracture)–immobile (matrix) mass transfer, not of equilibrium adsorption.

### Estimating total $CO_2$ storage capacity of the field site and upscaling potential

The total mass of $CO_2$ that could be mineralized in this system is challenging to estimate. Field observations suggest that the total extent of carbonation of a basalt is limited by the availability of pore space rather than the mass of available reactive rocks. This is because water–$CO_2$–basalt interaction leads to an increase in the overall volume of solids, reducing the pore space and blocking flow paths over time. Also, this process provokes the formation of numerous hydrous silicate minerals, including clay and zeolite minerals, as well as $CO_2$-storing carbonate minerals[45,46]. Overall, the volume of carbonate minerals produced by water–$CO_2$–basalt interaction may be only 20% or less of the total volume of precipitated secondary minerals[45]. This further limits the space available for mineral carbonation reactions in basalts. The overall efficiency of mineral carbonation can vary by adjusting the fluid injection rates and the composition of this injected fluid, including $CO_2$ concentration[45]. Each of these factors could be varied during a subsurface mineral carbon storage effort. If all of the effective flow volume was available for just calcite precipitation, the available 24,000–43,000 $m^3$ of effective pore space in our pilot system would be sufficient to contain 22,000–40,000 tons of mineralized $CO_2$ in total. This value is largely an overestimate owing to the likely formation of hydrous silicate secondary phases, which will consume some of the available pore space. Also, it is unlikely that all of this pore space would be available for secondary mineral precipitation, as fluid pathways

may be blocked. Some of these challenges and limitations might be overcome by fracking the subsurface rock formation[47], but this opinion has been relatively unexplored so far in basaltic systems.

One potential advantage of using the approach of this study for subsurface storage is that it limits the consumption of energy needed for subsurface storage. Energy is required to pump water from the subsurface, for which the amount of energy required is a function of the depth of the water table, flow rate and the efficiency of the pump. The injection of $CO_2$, using our approach, requires $CO_2$ to be delivered at a pressure that is high enough to exceed the hydrostatic pressure at its delivery depth in the wellbore. With further pressure that could be needed to enhance $CO_2$–water mixing by means of a static mixer or a nozzle, the total $CO_2$ delivery pressure at the surface would be around 12–14 bar. This range is 8–16 times lower than what is typically needed for conventional CCS. The generated water-charged $CO_2$ injection is essentially driven into the system by gravity. Thus, the overall energy penalty by making use of our $CO_2$ storage approach may be less than for conventional storage requiring injection of fluids into deeper and higher-pressure systems. Nevertheless, this proposed technology is seen as complementary to conventional CCS and not an alternative, as the geological conditions dominate.

## Data availability

The supplementary data files contain all data shown in the figures, including measured production well pH, fluid concentrations and the isotopic compositions of the oxygen and carbon recovered from the broken submersible pump. These data files are openly available from the Mendeley Data public repository 'Jizan $CO_2$ mineralization pilot in Saudi Arabia', Version 2, https://doi.org/10.17632/xh2d3cxggx.2.

## Code availability

The PHREEQC code used to calculate the saturation states of fluids in this study is freely available at https://www.usgs.gov/software/phreeqc-version-3. The Kinec_v3 database used is freely available at https://github.com/Mou1a/Kinect_V3. The input files used for calculating the saturation indexes are available from the Mendeley Data public repository 'Jizan CO2 mineralization pilot in Saudi Arabia', Version 2, https://doi.org/10.17632/xh2d3cxggx.2.

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

**Acknowledgements** We thank Saudi Aramco for conceiving and conducting this pilot project and for providing substantial financial support. We also thank King Abdullah University of Science and Technology (KAUST) Research Funding Office for additional support under award no. 4357. We thank KAUST research publication services and T. Leach, scientific illustrator, for producing Fig. 1a. We are grateful to C. Ballentine and R. Haese for insightful comments leading to an improved final manuscript.

**Author contributions** S.A., Z.A. and N.K. conceived the project. E.H.O., S.A., S.R.G. and G.B. designed the research. J.F., N.M., D.B. and T.F. mapped and interpreted subsurface fault patterns. H.H., M.A. and G.B. interpreted and modelled subsurface fluid flow and performed research. M.A. and A.O. contributed new analytic tools and measurements. E.H.O., S.A., M.A., H.H. and A.O. analysed the data. A.A. interpreted field observations. E.H.O., S.A., M.A., A.A. and H.H. interpreted the data and wrote the paper, with intellectual input from G.B., Z.A., A.A., A.O. and S.R.G. M.M. performed and interpreted XRD spectra.

**Competing interests** The authors declare no competing interests.

**Additional information**
**Correspondence and requests for materials** should be addressed to Eric H. Oelkers.

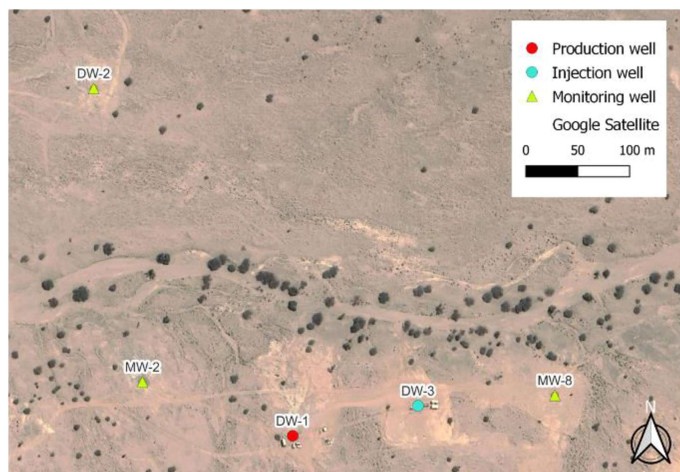

**Extended Data Fig. 1 | Map showing the relative locations of the five wells drilled as part of this study.** The names of the wells are provided in the figure. Aerial image ©2019 Airbus.

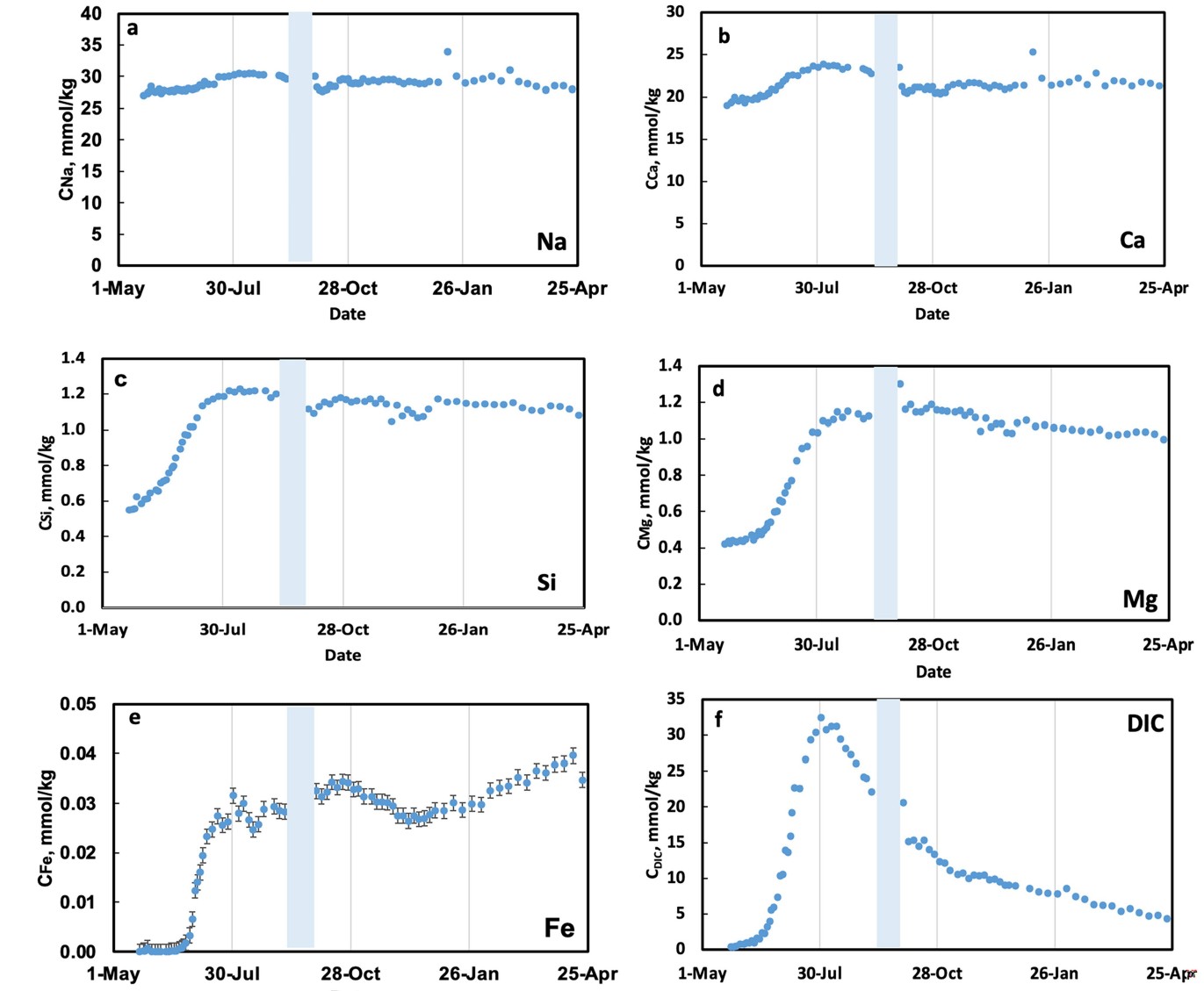

**Extended Data Fig. 2 | Measured compositions of the indicated elements in collected production well fluids from 22 May 2023 until 21 April 2024.** The analytical uncertainties of the measurements are smaller than the symbol size, with the exception of Fe, for which error bars are provided. The DIC concentration, presented in the bottom-right plot, was determined from measured in situ pH, alkalinity, temperature and total dissolved element concentrations using PHREEQC[36] together with its Kinec_v3 database[23].

The shaded region of each plot indicates the time when the production well's submersible pump failed, resulting in a halt of the fluid circulation for approximately 18 days. Despite this interruption, the consistency observed in the trends of the measured data after circulation resumed suggests that the shutdown period had no notable impact on the functioning of the subsurface system.

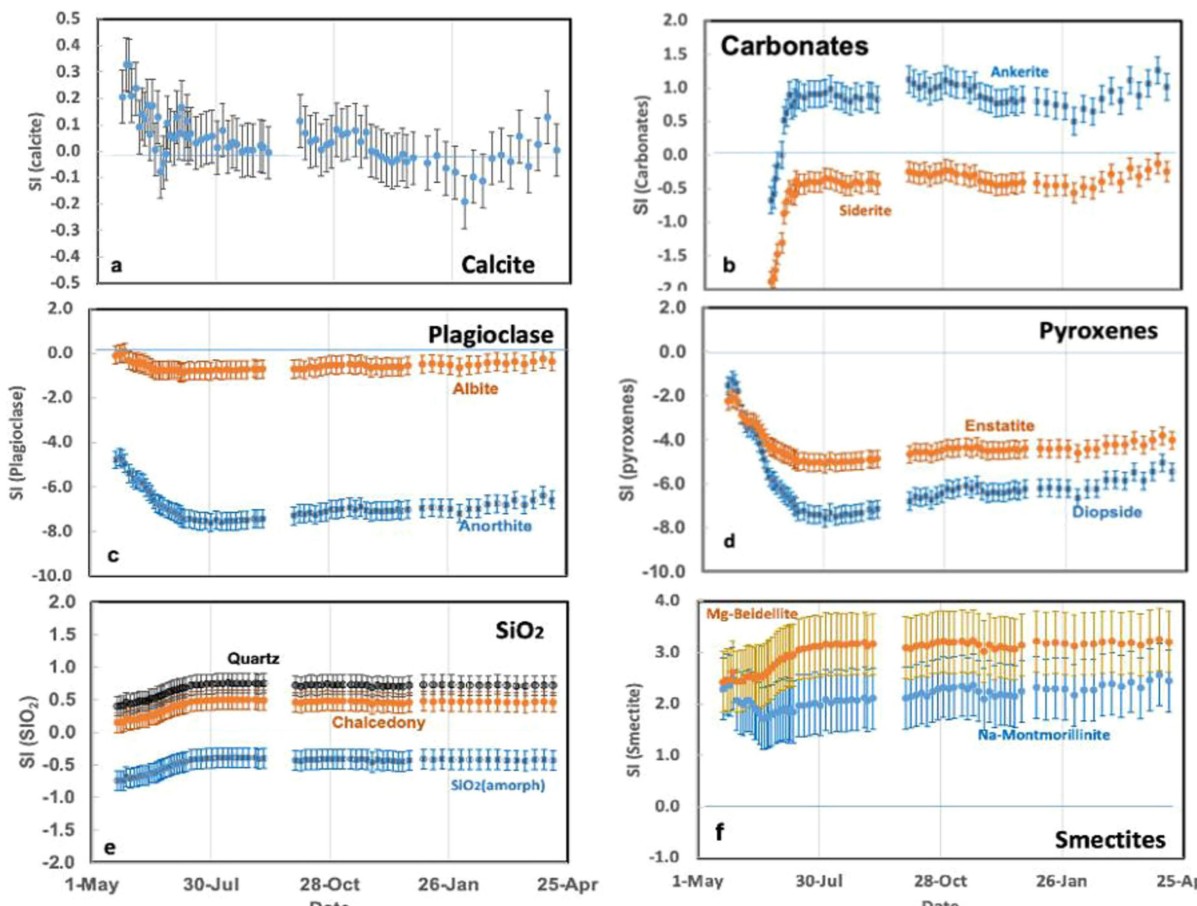

**Extended Data Fig. 3 | Saturation states of the collected production well fluids from 20 May 2023 until 21 April 2024.** These values are calculated for the in situ temperature of 45.5 ± 1 °C. A positive saturation state implies that the fluid is supersaturated with respect to the indicated mineral, a saturation state of zero indicates that the fluid is in equilibrium with respect to the mineral phase and a negative saturation state indicates that the fluid is undersaturated. These saturation states were calculated using PHREEQC[36] together with its Kinec_v3 database[23]. Note the gap in the dataset during September is because of a failure in the production well's submersible pump, which resulted in a halt of the fluid circulation for approximately 18 days. Despite this interruption, the consistency observed in the trends of the measured data after circulation resumed suggests that the shutdown period had no notable impact on the functioning of the subsurface system. The error bars correspond to an estimated uncertainty in the saturation indexes of calcite equal to ±0.1, siderite, quartz, chalcedony and amorphous $SiO_2$ equal to ±0.15, for the feldspars and pyroxenes equal to ±0.4 and for the clay minerals equal to ±0.6.

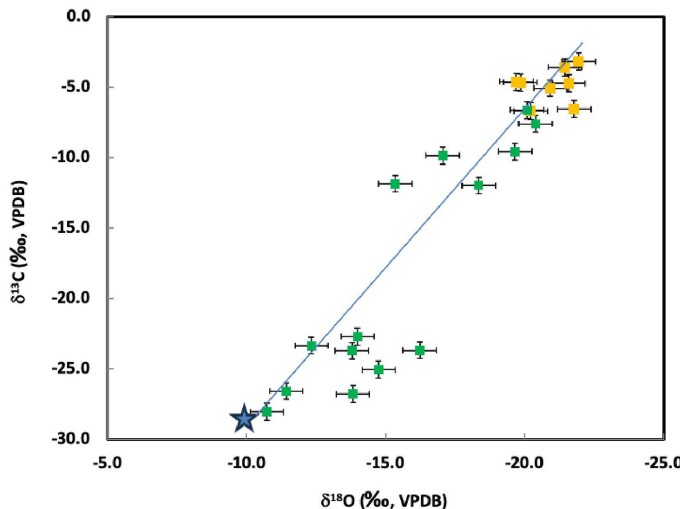

**Extended Data Fig. 4 | Isotopic compositions of carbon as a function of the corresponding oxygen isotopic composition of samples collected from the Jizan pilot project production well.** The yellow symbols represent samples collected from veins of well cuttings collected before the injection of $CO_2$ and the green symbols represent the compositions of carbonates recovered from the submersible pump. The blue star shows the isotopic compositions of carbon and oxygen in equilibrium with the injected $CO_2$ and the recirculating water and the blue line traces an approximate mixing curve and the calcite in the subsurface rocks before the injection. The coherence between this blue line and the symbols is consistent with the recovered carbon to be a mixture of fresh carbonate and carbonate present in the subsurface before the $CO_2$ injection.

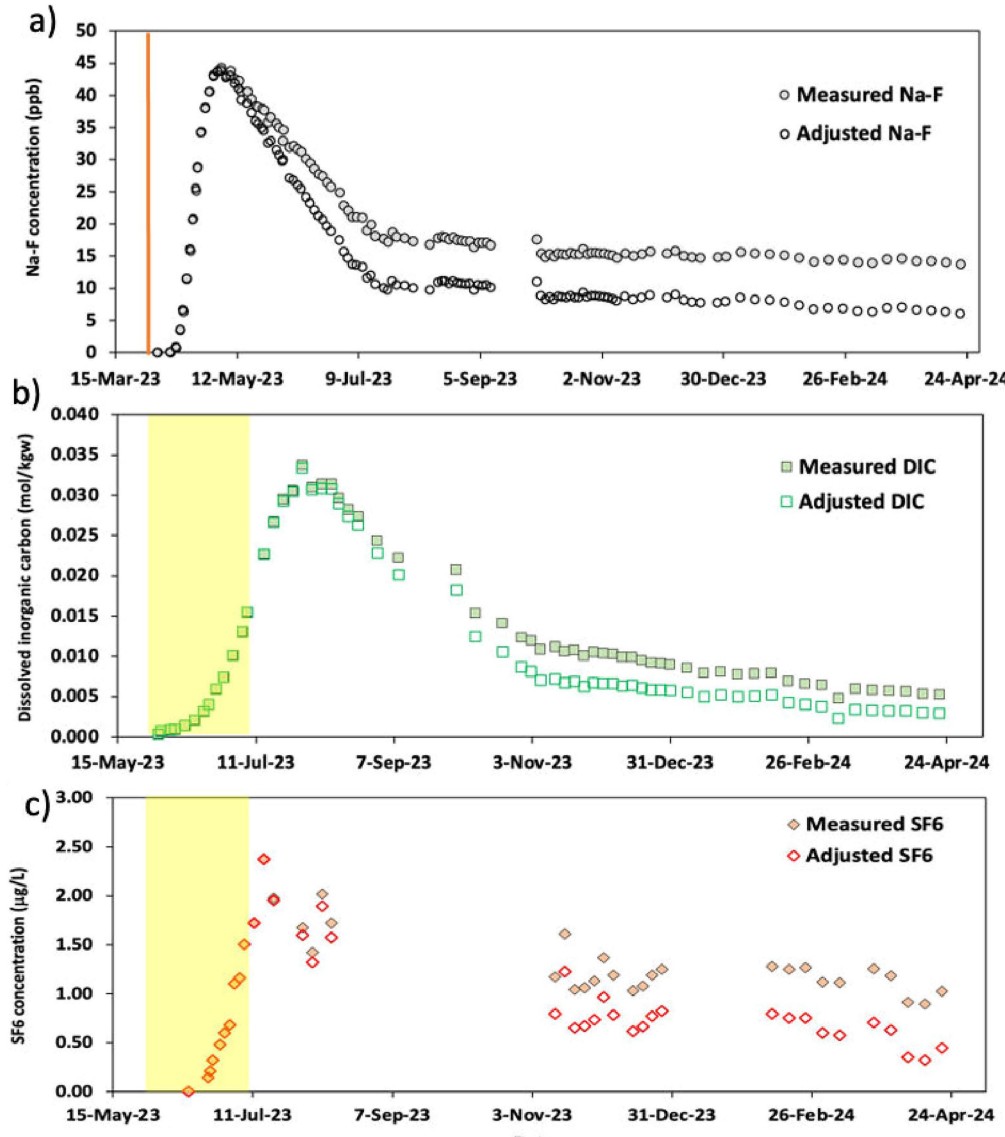

**Extended Data Fig. 5 | Comparison of measured concentrations of inorganic carbon and chemical tracers in production well fluids with those adjusted by deconvolution to remove the influence of recycling. a**, NaF concentrations. The vertical orange line marks 4 April, the date when 2.3 kg of NaF was introduced as a slug into the reinjected water. **b**, DIC. **c**, $SF_6$. Note that the date scale on **a** differs from that of **b** and **c**. The yellow shaded areas show the period during which $CO_2$ and $SF_6$ were injected into the system. The gap in the dataset between August and September 2023 could not be recovered owing to failure in the production well pump. Despite this interruption, the consistency observed in the trends of the measured data after circulation resumed suggests that the shutdown period had no notable impact on the functioning of the subsurface system. The analytical uncertainty in the measurements is approximately equal to the symbol size.

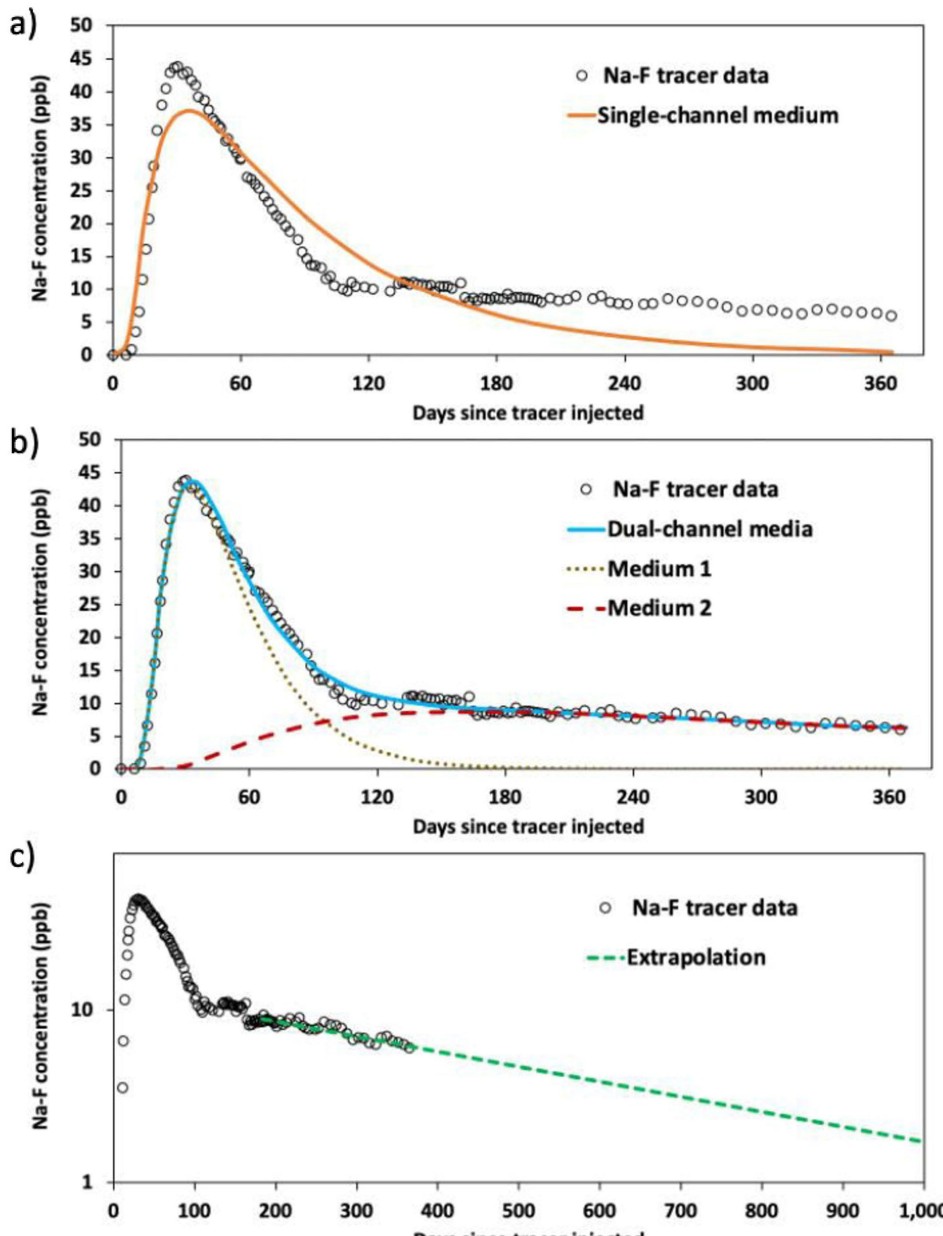

**Extended Data Fig. 6 | Numerical analysis of the temporal evolution of NaF tracer concentrations recovered from the production well. a**, Comparison of measured NaF concentration with a single-channel simulation model showing a poor match, highlighting the necessity for a dual-channel model. **b**, An improved match using a dual-flow system of high-permeability and low-permeability media, emphasizing the complex heterogeneous nature of the Jizan Group basalts. **c**, Extrapolation of NaF concentration data using an exponential decline to estimate the effective pore volume encountered by the tracer. The measured data align as a straight line when plotted on a semi-log scale. The analytical uncertainty in the measurements is approximately equal to or less than the symbol size.

**Extended Data Table 1 | Oxide composition in weight percent of the cuttings collected from the injection well, as measured by XRF**

| Formula | Concentration (%) | Stat. error (%) |
|---|---|---|
| $SiO_2$ | 47.08 | 0.75 |
| $Al_2O_3$ | 13.57 | 0.74 |
| $Fe_2O_3$ | 13.1 | 1.5 |
| CaO | 8.85 | 1.0 |
| MgO | 6.88 | 1.25 |
| $Na_2O$ | 2.46 | 0.38 |
| $TiO_2$ | 1.55 | 0.29 |
| $K_2O$ | 0.69 | 0.34 |
| $Mn_2O_3$ | 0.21 | 0.02 |
| $P_2O_5$ | 0.22 | 0.05 |
| LOI* | 5.07 | 1.57 |

The reported values are the average and the statistical error, taken as twice the standard deviation of ten samples collected from depths ranging between 255 and 400 mbs.

*Loss on ignition.

**Extended Data Table 2 | Summary of the mineral composition of the injection well cuttings, as determined by XRD combined with Rietveld refinement**

| Phase | Average Weight Percent | Standard Deviation |
|---|---|---|
| Plagioclase (Andesine) | 36.3 | 12.2 |
| Pyroxene (Augite) | 25.7 | 5.6 |
| Amphibole (Richterite) | 2.6 | 2.5 |
| Chlorite (Clinochlore) | 11.6 | 4.06 |
| Smectite | 6.1 | 2.5 |
| Quartz | 5.3 | 2.6 |
| Zeolite (Laumontite) | 11.3 | 12.4 |
| Calcite | 1.0 | 1.9 |

The reported values are the average and standard deviation of ten samples collected from depths ranging between 255 and 400 mbs.

**Extended Data Table 3 | Mineralogical and isotopic compositions of 14 solid samples collected from inside the damaged submersible pump on 10 September 2023 as determined using XRD and Rietveld refinement**

| Sample | Quartz | Plagioclase | K-spar | Biotite | Amphibole | Pyroxene | Chlorite |
|---|---|---|---|---|---|---|---|
| 9-Carb* | | | | | | | |
| 9-A | 16 | 33 | - | 5 | 8 | - | 20 |
| 9-B | 2 | 34 | - | - | - | 31 | 24 |
| 9-D | 2 | 24 | 16 | - | 9 | 24 | 17 |
| 9-E | 17 | 16 | 12 | - | 9 | 23 | 18 |
| 9-F | 56 | 32 | - | - | - | 6 | - |
| 9-G | 4 | 29 | 12 | - | 9 | 19 | 22 |
| 9-H | 38 | 14 | 17 | 4 | 7 | 5 | 9 |
| 9-J | 50 | 16 | - | 7 | 7 | 4 | 8 |
| 9-K | 36 | 22 | 11 | 5 | 5 | 4 | 8 |
| 9-L | 38 | 13 | 10 | 6 | 12 | 4 | 7 |
| 9-M | 32 | 14 | 19 | 6 | 8 | 5 | 8 |
| 9-N | 30 | - | - | 7 | - | - | - |
| 9-O | 10 | 7 | - | - | - | - | - |

| Sample | Calcite | Siderite | Ankerite | Magnetite | Goethite | $\delta^{13}C$ | $\delta^{18}O$ |
|---|---|---|---|---|---|---|---|
| 9-Carb* | | | | | | -26.8 | -13.8 |
| 9-A | 4 | 3 | 2 | 3 | 3 | -11.9 | -15.3 |
| 9-B | 9 | - | - | - | - | -9.9 | -17.1 |
| 9-D | 1 | - | - | - | | -7.6 | -20.4 |
| 9-E | 2 | - | - | - | 1 | -9.6 | -19.6 |
| 9-F | 1 | - | 3 | - | 2 | -12.0 | -18.3 |
| 9-G | 2 | - | - | - | - | -6.6 | -20.1 |
| 9-H | 3 | 1 | 1 | - | 2 | -22.7 | -14.0 |
| 9-J | 2 | 1 | 1 | - | 2 | -25.0 | -14.7 |
| 9-K | 2 | 1 | 2 | - | 1 | -23.3 | -12.3 |
| 9-L | 2 | 1 | 1 | - | 2 | -23.7 | -16.2 |
| 9-M | 2 | 1 | 1 | - | 2 | -23.7 | -13.8 |
| 9-N | 10 | 4 | 1 | 25 | 12 | -28.0 | -10.7 |
| 9-O | 14 | 2 | 2 | 40 | 14 | -26.6 | -11.4 |

Mineral compositions are provided in weight percent. Note that XRD identifies the minerals present through their structure and does not provide the exact chemical composition of the identified minerals. The XRD patterns of all samples are provided in Supplementary Fig.1. *This sample consisted of carbonate dust that fell off the pump as it dried.