## [Peer Review file · Nature]

CO₂ mineral storage in subsurface basalts by its co-injection with recirculating water

Corresponding Author: Professor Eric Oelkers

Version 0:

Reviewer comments:

Referee #1

(Remarks to the Author)

This study reports data from a pilot scale experiment on subsurface CO₂ sequestration.

The outstanding feature of this work is that this is the (to my knowledge) first reported pilot scale CO₂ injection using recirculated water. Analysis of carbonation extent using tracers and XRD analysis of solid samples recovered from the broken pump are very valuable to the scientific community.

- **Validity:** Does the manuscript have flaws which should prohibit its publication? If so, please provide details.

While the fact that the solid materials were recovered due to a broken pump makes is less intentional and the analysis of the solids (XRD) is not very comprehensive; these are not flaws that prohibit publication. Due to the urgent nature of the topic, I think it's a good idea to have XRD results in this manuscript and hopefully more comprehensive solid analysis using microscopy in a follow-on study.

- **Originality and significance:** If the conclusions are not original, please provide relevant references. On a more subjective note, do you feel that the results presented are of immediate interest to many people in your own discipline, and/or to people from several disciplines?

To my knowledge, this is the first report of pilot project CO₂ subsurface injection with recirculated fluids. Therefore, given the pressing nature of atmospheric CO₂ increase and global warming, it is justified for this study to reach a broad audience. I therefore think that these results would be of interest to many people from my own discipline and to people from other disciplines as well.

- **Data & methodology:**

There are several points in the methods, where data reporting and methodology could be improved. Here are specific requests:

- 1) Statistics: reservoir temperature of 46°C – what is the error for this temperature?
- 2) Clean PET bottles – how were these bottles cleaned?
- 3) Line 392 - pre-cleaned 1.0 L glass bottles – how were these glass bottles pre-cleaned?
- 4) What is the purity of the HNO₃ used? Which wavelength were chosen the elements analyzed by the ICP-OES analysis?
- 5) While it is somewhat visible from the legend in the Figures in the SI which datafiles were used for identification of the mineral phases, it would be easier for any reader if the authors could provide a table or something similar with references of which crystallography file they used for which phase.
- 6) Statistics: the Fig S 4 $\delta^{18}\text{O}/\delta^{13}\text{C}$ measurements: Is it possible for the authors to add error bars? They are provided in the measurement section but it would be helpful for the reader to see them here.
- 7) Figure 3: Are the error bars also smaller than the symbols here similar as Figure 2?
- 8) Figure S1: can the part of the graph in which the pump failed maybe highlighted?
- 9) In the concentration of Na, Ca, Si and Mg, are some datapoints that are laying on the x-axis as 0 – is this correct or are these below detection limit? If it's below detection limit, can these measurements be taken out?

- **Conclusions:** Do you find that the conclusions and data interpretation are robust, valid and reliable?

The authors state that Mg is incorporated into ankerite and siderite – could there be also some Mg-carbonate phases forming? What solubility products does the database used for the PHREEQC modeling have for these minerals? Maybe it need to be adjusted? Do the authors have any microscopy evidence of the Mg being incorporated into the ankerite and siderite?

• Suggested improvements: Please list additional experiments or data that could help strengthening the work in a revision. I suggest that the authors provide the associated data and the PHREEQC code used to calculate the supersaturation in an data repository. Maybe this is available but I haven't found it.

Fig. S3 – There seem to be some unidentified peaks – could the authors discuss these more? In particular at ~37 degree 2theta – the peak is very large compared to what would be expected based on reference data.

Same peak around 37 is visible in Fig. S2, but here microcline is added (green), however peaks around 37 does not seem to match the observed spectra.

Line 623: There also seems to be double labeling of some figures, e.g. Fig. S3 Sample 9-F has an additional Figure caption.

Line 736, Table S4: The database named for PHREEQC here is different than in the methods section. Same for table S5.

Figure S2, line 589 – several issues with this figure:

-how was the SI with regard to “carbonates” calculated?

-Does this take into account any amorphous carbonates that could form?

-Why is SI calcite not on the same graph as the other carbonates?

-Also labeling of the subfigures is missing for Figure S2.

-I think the label for the plagioclase graph y-axis is wrong, should it be SI plagioclase?

Also the graph can be difficult for color blind people to read, instead of just using two different colors, can different symbols be used?

• References: Does this manuscript reference previous literature appropriately? If not, what references should be included or excluded?

This is okay.

• Clarity and context: Is the abstract clear, accessible? Are abstract, introduction and conclusions appropriate?

I think parts of the manuscript could be improved to address non experts, e.g. non geologists since this is a broad audience journal. One suggestion is explaining the unit mbs (meter below surface) since this is likely not known for a non geology audience.

In the abstract, the second sentence seem to be missing some words:

“A major challenge to applying this process at scale as it can require 20 to 50 times or more water than the mass of CO₂ stored.” I suggest rephrasing it.

Kinetic.v3 database? Is this the right database? Why chosen?

Line 80/81: Water circulation in this system was isolated from the atmosphere to avoid potential CO₂ leakage and to prevent the addition of O₂ to the subsurface.

Can the authors add why addition of O₂ to the subsurface has to be avoided?

The unit for the piezometric level (mbs) – since this is a broad audience journal, I suggest to spell this unit out the first time.

Line 92:

NaF as abbreviation for sodium fluorescein is somewhat misleading due to the chemical formulat for sodium fluoride also being NaF. I suggest that the authors use a different abbreviation.

Referee #2

(Remarks to the Author)

This study demonstrates for the first time CO₂-water co-injection and permanent carbon storage in fractured basalts without the use of external water in a field experiment. This is particularly important in regions where basalt formations are widespread, and water resources are limited. The field study uses an injection and a production well where injected water is constantly enriched with CO₂ and the produced water is re-injected into the injection well avoiding any external water. The study uses water composition of the produced water and mineral analysis to constrain the fate of injected CO₂ in the subsurface.

The characterisation of the subsurface hydrology is complicated by the fracture network where fractures serve as primary fluid flow pathways. The tracer NaF was injected for about a day and its concentration at the production well was used to constrain the reservoir hydraulic properties. A dual-channel model was successful in matching the observed changes in tracer concentrations at the production well. Fractures were estimated to contribute to about 90% of flow while the matrix rock contributed about 10% of flow. The tracer SF₆ was added to the injection water at a constant concentration throughout the operation (about one year) and was used to estimate carbon mineralization.

My main concern with this study is an insufficient and probably incorrect characterisation of the subsurface hydraulic regime:

Firstly, I have little confidence in the use of NaF at elevated temperature for subsurface flow characterisation given the study by Black et al. (2017, doi.org/10.1016/j.ijggc.2017.10.012.)

Secondly, it is not clear (not explained) why SF₆ concentrations decline over time at the production well given it is continuously added to the injection water. What is the sink for SF₆? It may be related to the two possible processes which come to mind:

The water level at the production well dropped by over 150 m relative to the pre-operation level. This means a deep (radial) cone of depression has formed drawing in water from all sides. Water from outside of the plume coming from the injection

well would have contributed significantly to the chemical composition of samples; I expect significant dilution was occurring. While dilution is mentioned in principle (p. 4, row 30, p. 5, row 21), it has not been quantified, and the proportion of 'recirculated' water remains unknown.

Secondly, a potential sink for the tracers and indeed any components enriched at the injection well (DIC, H⁺, liquid tracers) could be molecular diffusion across the fracture – matrix rock interface. As fractures are by far the most dominant fluid flow pathway ions with a concentration higher than in the matrix rock would diffuse into the matrix rock and may (H⁺, DIC) or may not (inert tracers) react with minerals. Several studies have characterised transport and reactive transport across such lithological interfaces. For example, Gilmore et al. (2020, doi.org/10.1029/2020GL087001) quantified molecular diffusion of dissolved CO₂ across such a lithological interface and Phukan et al. (2021, doi.org/10.1029/2021WR030275) showed concurrent mineral dissolution and precipitation in matrix rock adjacent to preferential flow paths of CO₂-saturated water in basalt in an experiment.

I am not convinced the observed carbonate mineral content in the submersible pump is an indication of carbonate mineral precipitation in the reservoir. The pump lowers the water pressure creating a pressure gradient from the outside to the inside of the well. This could lead to CO₂ degassing and shifting the carbonate equilibrium towards a regime where carbonate minerals precipitate. Mineral precipitation (scale formation) is often observed in oil & gas as well as in geothermal production wells for the same reason.

Importantly, the main conclusion from this study remains the same when taking my main concerns (above) into account: The study still demonstrates the opportunity to co-inject CO₂ and water into fractured basalts without the use of external water. It places greater emphasis on water dilution and molecular diffusion of DIC into the matrix rocks than on carbonate mineral precipitation.

Minor comments:

Title: I think the term 'fractured basalts' should be in the title given the importance of fractures for CO₂ storage in this scenario. An alternative title could be: Recirculating water for CO₂-water injection into fractured basalts.

Abstract, second sentence: Incomplete sentence (verb is missing) and change '20 to 50 time or more' to '20 to more than 50 times'.

p. 2, line 12: It is my understanding that desalination facilities do not produced CO₂ emissions themselves (Scope 1 emissions). Consider removing / replacing 'desalination facilities' from the list.

p. 2, line 28: What is / Is anything known about burial and uplift over the last 20 to 30 million years?

p. 2, line. 31: How far apart were the three wells with no hydraulic connectivity to DW-1 and DW-3? It is an indication of the heterogeneity in the sub-surface which is relevant to the proposed technology. Could the location of the three wells be included in Fig. 1?

p. 4, l. 3 – 6: The primary orientation of the fractures and faults is (sub-)vertical based on Figure 1 and you suggest most of the transport occurs within faults. Comment on the importance of (sub-)vertical flow vs horizontal flow, if possible. You may also want to emphasize horizontal faults and fractures more strongly in Figure 1.

p. 4, l. 19: Include here: 'Dissolved inorganic carbon (DIC) was calculated based on alkalinity and pH.' Yes, I know you mention this in the Method section, but it is important to introduce DIC early in the ms.

p. 5, l. 9: Primary minerals are typically close to equilibrium in undisturbed subsurface systems. Why are primary plagioclase and pyroxene strongly undersaturated before the commencement of CO₂ injection?

p. 5, l. 12 – 15. Long and convoluted sentence. Make two sentences out of it. Secondly, calcite saturation at the production well only suggests that a re-equilibration has occurred. It does not tell us how this occurred. It could be through fluid mixing as well.

p. 5, l. 31: include zeolites as possible secondary minerals, see Lu et al., <https://doi.org/10.1016/j.earscirev.2024.104813>

p. 7, l. 1 – 2: I have little confidence in the carbon mineralization estimate given my concern NaF may not be conservative at elevated temperatures and the decline in SF₆ concentration at the production well remains unexplained, see my comments earlier.

p. 9, l. 28: Why was diaspore chosen for the equilibrium with Al(aq)? Diaspore was not identified as a primary mineral. I suggest using clinocllore or laumontite.

p. 11, l. 18 – 21: Calculate how much of the initial mass of NaF (2.3 kg) was recovered at the production well. Comment on potential processes resulting in the loss of NaF, see also major comment earlier.

Referee #3

(Remarks to the Author)

The ability to capture CO₂ within mafic and ultramafic rock bodies through mineralisation of the carbon is an important CCS technology. Here, the authors make the case for a successful demonstration/pilot of a process, which re-circulates the water in which the CO₂ is dissolved rather than a single one-way pumping of water into the rock body. Fresh water use in many technologies competes with domestic need and is becoming more and more critical in the consideration of implementing them, particularly in regions where water supply is already stressed. This is an important paper and something that I hope Nature considers publishing.

This paper is important for several reasons. This is a demonstration at scale. While modest, the mass of 131 tons injected CO₂ is significant and the injection/receiver well conditions and scale (250m depth and 130m distance) representative of real world systems rather than idealised experiments. The data quality and presentation is strong. There is little uncertainty that 70% of the CO₂ was reacted within the rock; the supporting NaF tracer, chemical, mineralogical and isotopic data is robust and well documented. The demonstration of primary vs secondary porosity and their respective character and proportion was particularly nice to see.

I am nevertheless, left feeling unsatisfied with the conclusions drawn. It is as if showing that 70% CO₂ removal is enough (it is good) and this alone paves the way for a successful demonstration of concept. I would really like the authors to consider the following questions, because I think they have the data (or should have) to simply add these to the publication:

What is the estimated change of secondary or total porosity from the dissolved mineral and precipitated mineral mass balance. This can be used to derive a crude lifetime and total carbon capacity for the trial system (acknowledging heterogeneity of both precipitation and primary vs secondary porosity response). Does it clog up next week or will this keep storing CO₂ for years....? How might this scale up further and how reliant is this on the secondary porosity?

What is the power consumption of recirculation? How does this compare with one way pumping? This addresses cost and practical difficulties that may or may not make this process feasible beyond the water volumes used?

The observation that several proximal wells were not in hydrogeological connection with the pumping well left me wondering what level of conditioning (e.g. fracking) may be required to roll out this type of technique and the practicalities involved. Is this practical?

None of the questions above need be addressed in any great detail, but should be acknowledged to provide more depth to an important paper and better represent the challenges (or ease) ahead for this style of CCS technology.

Chris Ballentine

Version 1:

Reviewer comments:

Referee #1

(Remarks to the Author)

The authors have addressed all requested changes and answered questions from reviewer.

The main point which I find somewhat problematic is the method used to clean the bottles.

So what I understand is that the bottles contained drinking water + were pre-cleaned with production well water. This is somewhat problematic in my opinion because there are various anions+ cations both in the original drinking water as well as in the well water that can adsorb to the bottle walls. I don't think that the ICP-OES data can be trusted that much then. The ICP-OES measurements for the low concentration elements (Fe, Mg, Si) could potentially be influenced by this cleaning method.

However, I think it should be referenced and then it should be okay since the other elements are in much larger concentrations. Also, there is some small other changes that would be good to make:

What dissolution ratio was used for ICP-OES acidification? The acid concentration that the authors state is 2%, which is usually the end concentration.

Please also change in Figure captions and Figure labelling as well (e.g. Figure 2; Extended data figure S5) from NaF to Na-F.

I also feel that it would be beneficial to include the information in the text that the desalination plants in Saudi Arabia are large CO₂ emitters - this is explained in the comments to the reviewers but it's not clear how the reader is supposed to know that if it's a local specialty.

Overall, since this is an important study providing the industrial pilot scale data, I think it makes sense that the manuscript will be published after these corrections are made and therefore recommend minor revisions.

Referee #2

(Remarks to the Author)

Main comments / concerns

1. This reviewer questions the use of Na-F at elevated temperature for subsurface flow characterization given the study by Black et al. (2017, doi.org/10.1016/j.ijggc.2017.10.012.). The paper referred to (Black et al., 2017) states an apparent fluorescein loss caused by adsorption to mineral surfaces in sandstone dominated reservoirs at comparatively low reservoir pH and moderate T. However, the authors themselves expressed doubts about the validity of their findings, since most measured fluorescein concentrations from muds and natural water samples they tested are for the most part statistically indistinguishable between the two groups (within $\pm 2s$). Moreover, we recovered 50% of our injected Na-F during our study period with additional Na-F projected to be recovered after this time. This suggests that any adsorption would be negligible in our study. We have explained this in text added to the revised Methods Section.

Response: The Black et al. 2017 paper reports on two field experiments where fluorescein was used in deeper reservoirs. According to that paper, fluorescence results were "difficult to interpret" in one experiment, while the large drop (~90%) in fluorescence concentration in back-produced water in the second experiment "indicates there is a problem with using the fluorescein tracer in acidic CO₂ saturated waters (the predicted pH of injection waters from Test 2 being 4.5), which may be due to adsorption to mineral surfaces such as quartz (Moola et al., 2014)". Non-conservative behaviour of fluorescein is also stated in the Abstract and in the Conclusion sections.

2. This reviewer asks why SF₆ concentrations decline over time at the production well given it is continuously added to the injection water and asks what is the sink for SF₆? This is because of the continuous dilution of the subsurface water in our studied system. The water level at the production well was lowered by 150 meters, This decrease draws water in from the surrounding subsurface over time. We have added text to the methods section of the revised paper to inform this to the reader.

Response: There may be a misunderstanding here. My question refers to the following conditions: If SF₆ is continuously added to the inflow water and the inflow water has a constant concentration of X, why is the concentration in the outflow (produced) water declining over time? If SF₆ behaves conservative and water dilution is constant, the outflow concentration should remain the same.

3. This reviewer expresses concern that the carbonate precipitated on the submersible pump was due to a water pressure prop leading to CO₂ degassing, shifting the carbonate equilibrium towards a regime where carbonate minerals precipitate. We disagree strongly with this suggestion. The partial pressure of dissolved CO₂ was always less than 1 bar in the fluid phase. The pressure in the pump was not less than 5 bars. The exsolution of the fluid is therefore not favored and very unlikely. Moreover, no CO₂ gas was observed in the recirculated fluid at the injection well that included a gas trap and a CO₂ gas detector at the wellhead. We have added text to the end of the Identification of solids collected from the damaged submersible pump section of the methods section of the revised manuscript to explain this to the reader.

Response: OK, if CO₂(g) was monitored in the production well and it was not detected, that's fine. Still surprises me.

4. Response: The authors did not respond to another major concern I have, namely the potential diffusion from the fracture to the adjacent matrix rock for components enriched in the injection well (DIC, H⁺, liquid tracers). I still think this is a critically important question as it could mean that the loss of dissolved CO₂ is (to some degree) controlled by the transport away from the main fluid flow pathways and not by carbonate precipitation. This is also relevant to my first specific comment.

Specific comments

1. This reviewer suggests adding the term 'fractured basalts' to the title given the importance of fractures for CO₂ storage in this scenario. This reviewer suggests the alternative title: Recirculating water for CO₂-water injection into fractured basalts. We disagree. The main breakthrough of this study is the use of recirculating water and the success of mineral storage in old and altered basalts. The approach should work just as well in porous basalts. As such we have not altered the manuscript in response to this suggestion.

Response: See above.

2. This reviewer suggests changing the second sentence of the abstract for improved clarity. We agree and have revised this sentence in the revised manuscript to improve its clarity.

Response: Thanks.

3. This reviewer suggests that desalination facilities do not produce CO₂ emissions themselves (Scope 1 emissions) on page 2 line 12 of the manuscript and considers removing these words. Perhaps true in most of the world, but in Saudi Arabia, it is common practice to use oil to run dual electricity- desalination plants, and these are large CO₂ emitters in this region. Thus, the original sentence is correct and we have not change this in the revised manuscript.

Response: Many thanks for the clarification.

4. This reviewer asks of there more information on the burial and uplift of the region over the past 20 to 30 million years on page 2, line 28. We agree and have added a sentence and a reference to inform the reader of the past geologic history of the region.

Response: Thanks.

5. This reviewer asks that we provide the distances of the wells that had no hydraulic connectivity to DW-1 and DW-3 on page 2, line. 31 and asks if the location of the three wells be included in Fig. 1? In response to this suggestion, we have

added a map showing the relative positions of all the wells drilled for this project as an extended data figure and referred to this in the caption of Figure 1.

Response: Thanks.

6. This reviewer suggests we emphasize the importance of the orientation of the horizontal faults and fractures more strongly in Figure 1 and in text on page 4, lines 3 to 6. We agree and have added a sentence to the revised manuscript to alert the reader to the existence of sub-horizontal fractures in this system.

Response: Thanks.

7. This reviewer asks that we describe how we determined DIC concentration on page 4 line 19. This was described in the methods section of the original manuscript. Consequently, we have not changed the manuscript in response to this comment.

Response: OK, thanks.

8. This reviewer asks for insight into why primary plagioclase and pyroxene strongly undersaturated before the commencement of CO₂ injection on page 5, line 9. We attribute this to the drawdown of fluids from near the surface to the production well. The pumping to the production well lowered the water level in the production well by 150 m. This resulted in the downward migration of cooler, shallower water regions down to the production well, leading these primary phases to be undersaturated in collected production well fluids. We have altered the text of page 5 of the revised manuscript to clarify this to the reader.

Response: Many thanks for the clarification.

9. This reviewer suggests we rewrite the sentence on page 5, lines 12 – 15 for improved clarity. We have altered this sentence somewhat in the revised manuscript, but we believe that this was and continues to be clear in the revised manuscript.

Response: OK.

10. This reviewer suggests we consider the role of zeolite formation as secondary minerals on page 5, line 31. These minerals are undersaturated in all production well fluids so unlikely to form. We have added a sentence to this part of the revised manuscript to alert this to the reader.

Response: OK.

11. This reviewer expresses concern about using the NaF to dissolved carbon ratio to estimate the quantity of the carbon mineralization page 7, lines 1 – 2: This reviewer is concerned that NaF may not be conservative and to provide further insight into the decline in SF₆ at the production well. We agree and have added a paragraph to the methods section discussing this issue. Notably more than half the Na-F was recovered during the monitoring period, and modelling suggests more would arrive in the future. Moreover there is no significant difference between the fraction of carbon mineralized by using Na-F and SF₆. This further supports our interpretation of the data. We have noted this last factor in new text added to the methods section.

Response: This comment relates to my general concern about the use of fluorescene as discussed earlier.

12. This reviewer asks why we chose diaspore to fix the concentration of aqueous Al on page 9 line 28. This reviewer suggests using clinocllore or laumontite instead. Diaspore was chosen as it is observed to precipitate rapidly during experiments and as clinocllore and laumontite are calculated to be undersaturated in compared to diaspore in sampled solutions. We have added text to the methods of the revised manuscript to alert this to the reader.

Response: Many thanks for the clarification.

13. This reviewer asks that we calculate how much of the initial mass of NaF was recovered at the production well on page 11, lines 18 – 21, and to use this information to comment on the potential loss of Na-F through absorption and diffusion into the matrix. We agree and have added a paragraph to the methods section providing this number and discussing the implications.

Response: Many thanks.

Referee #3

(Remarks to the Author)

The revised document provides the substantive evidence base, with sufficient detail, to support the conclusions reached. The original manuscript lacked the broader impact and context of this work. The latter is now included in the methods section under the subsection 'Estimating total CO₂ storage capacity of the field site and upscaling potential'. While this covers the shortfall in the original document, it is somewhat of a shame for this contextual information not to be in the main text (the discussion section would be suitable) or have some snippets in the abstracts/opening paragraph. The context of the results will make this work more immediately relevant to a broader audience. The authors have nevertheless, addressed my comments suitably, and whether this is given more prominence is really an editorial call. I look forward to seeing this in-press.

Referee #1:

This reviewer provides a number of insightful constructive comments:

1. This reviewer asks that we provide and uncertainty estimate of the temperature. *The measured temperature was 45.5°C with a 2 standard deviation uncertainty of 0.5°C. We have added this information to the revised manuscript.*
2. This reviewer asks how we cleaned the PET bottles used for sampling *We agree and have added this information to the revised manuscript.*
3. This reviewer asks how pre-cleaned glass bottles were cleaned near line 392 of the original manuscript. *We have added this information to the methods section of the revised manuscript.*
4. This reviewer asks us to define the purity of the HNO₃ used and the wavelengths were chosen to analyze the elements by the ICP-OES analysis. *We agree and have added the wavelengths used in for the ICP-OES analyses and the origin of the HNO₃ used to the methods section of the revised manuscript.*
5. This reviewer asks us to better clarify which datafiles were used for identification of the mineral phase by providing a table with references of the crystallography files used for identifying each phase. *We agree and have added a table to the supplementary material providing the powder diffraction reference files used to identify the minerals found in the X-ray diffraction patterns shown in Supplementary Figure 1.*
6. This reviewer asks us to provide error bars on Extended Data Figure 2. *We agree and these have been added to this figure in the revised manuscript.*
7. This reviewer asks if the error bars in Figure 3 are smaller than the symbols. *The analytical uncertainty in the measurements is approximately equal to the symbol size. We have added this information to the caption of this figure in the revised manuscript.*
8. This reviewer asks us to highlight the time when the pump failed in Extended Data Figure 2: *We agree and have added this information to the revised version of this figure.*
9. This reviewer asks that we inform the reader if the concentrations that are apparently plotted as zero were in fact at zero or below a detection limit. And asks if there are below detection limit, can these measurements be taken out? *These values were in fact missing data, mis-plotted by EXCEL. We have removed these misleading values from this figure in the revised manuscript.*
10. This reviewer asks if could there be also some Mg-carbonate phases forming and if we have any evidence that Mg is being incorporated into the ankerite and siderite? *We have no direct evidence of Mg carbonates forming and the Mg carbonate minerals magnesite and dolomite were undersaturated in all sampled production well fluids. We do not have an analysis of the elemental compositions of the pump siderite and ankerite, as these were minor phases in the recovered solids, but Mg is commonly found in these minerals as mentioned in the original text. As such we have not altered the manuscript in response to this suggestion.*
11. This reviewer suggests that we include the associated data and the PHREEQC code used to calculate the supersaturation in a data repository. *We agree and have added the PHREEQC scripts used in the study to the public access repository associated with this manuscript.*

12. This reviewer suggests that we left some peaks in Supplementary Figure 1 unidentified, in particular one at $\sim 37^\circ 2\theta$. The reviewer suggests we try to better identify these. *We disagree that there is a missing phase in this interpretation. The peaks near 37° are attributable to quartz and goethite. Attempts to add an additional phase to these diffractograms did not improve the quality of the fit. As such we have not altered the manuscript in response to this comment.*
13. Correct the double labeling of the figure on line 623 of the original manuscript, e.g. Fig. S3 Sample 9-F has an additional Figure caption. *We agree and have removed the misleading extra caption from this figure*
14. Correct the inconsistency in the name of PHREEQC on line 736, Table S4, Methods Section, Table S5. *We acknowledge this inconsistency and have corrected in it the revised manuscript.*
15. Correct various issues on Figure S2, line 589 – notably: ...
 - how was the SI with regard to “carbonates” calculated? *These were calculated using PHREEQC together with measured chemical analyses including pH and alkalinity. We have revised a sentence in the methods section to clarify this to the reader.*
 - Does this take into account any amorphous carbonates that could form? *The calculation of all saturation indexes shown are independent of one another. This the formation of amorphous carbonates nor any other phase will affect the saturation indexes shown in the figure. As a result, we have not changed the manuscript in response to this comment*
 - Why is SI calcite not on the same graph as the other carbonates? *This was done so that SI values of calcite could be seen at a larger scale? As such we have not changed the manuscript in response to this comment.*
 - Also labeling of the subfigures is missing for Figure S2. *In response, we have added labels to each of the subfigures in this figure.*
 - I think the label for the plagioclase graph y-axis is wrong, should it be SI plagioclase? *We agree and have corrected this error in the revised manuscript.*
16. This reviewer also notes that Figure S2 can be difficult for color blind people to read, suggesting that can different symbols be used? *We agree and have changed the shape of the symbols in this figure in the revised manuscript.*
17. This reviewer suggests we can define the unit mbs (meters below surface) since this is likely not known for a non-geology audience. *This abbreviation was defined on line 29, page 2 of the original manuscript. As such we have not altered the manuscript in response to this comment.*
18. This reviewer suggests rephrasing the second sentence of the abstract for improved clarity. *We agree and have corrected this sentence in the revised manuscript.*
19. This reviewer asks why the Kenetic.v3 database was chosen for the calculation. *Kinec.v3 the most recently updated and documented database for PHREEQC. We have added text to the revised manuscript to alert this to the reader.*
20. This reviewer asks why we limited O₂ addition to the subsurface system in line 80/81. *This was done to avoid iron oxidation and/or microbial activity that could block subsurface flowpaths. We have added a sentence to the revised manuscript to alert this to the reader.*

21. This reviewer again suggests we define the units mbs for improved clarity. *This abbreviation was defined on line 29, page 2 of the original manuscript. As such we have not altered the manuscript in response to this comment.*
22. This reviewer suggests using a different abbreviation for sodium fluorescein as NaF could also refer to NaF. *This is a reasonable suggestion. In response to this we have changed this abbreviation to Na-F throughout the revised manuscript.*

Referee #2:

General Comments:

1. This reviewer questions the use of Na-F at elevated temperature for subsurface flow characterization given the study by Black et al. (2017, doi.org/10.1016/j.ijggc.2017.10.012). *The paper referred to (Black et al., 2017) states an apparent fluorescein loss caused by adsorption to mineral surfaces in sandstone dominated reservoirs at comparatively low reservoir pH and moderate T. However, the authors themselves expressed doubts about the validity of their findings, since most measured fluorescein concentrations from muds and natural water samples they tested are for the most part statistically indistinguishable between the two groups (within $\pm 2s$). Moreover, we recovered 50% of our injected Na-F during our study period with additional Na-F projected to be recovered after this time. This suggests that any adsorption would be negligible in our study. We have explained this in text added to the revised Methods Section.*
2. This reviewer asks why SF₆ concentrations decline over time at the production well given it is continuously added to the injection water and asks what is the sink for SF₆? *This is because of the continuous dilution of the subsurface water in our studied system. The water level at the production well was lowered by 150 meters, This decrease draws water in from the surrounding subsurface over time. We have added text to the methods section of the revised paper to inform this to the reader.*
3. This reviewer expresses concern that the carbonate precipitated on the submersible pump was due to a water pressure prop leading to CO₂ degassing, shifting the carbonate equilibrium towards a regime where carbonate minerals precipitate. We disagree strongly with this suggestion. *The partial pressure of dissolved CO₂ was always less than 1 bar in the fluid phase. The pressure in the pump was not less than 5 bars. The exsolution of the fluid in therefore not favored and very unlikely. Moreover, no CO₂ gas was observed in the recirculated fluid at the injection well that included a gas trap and a CO₂ gas detector at the wellhead. We have added text to the end of the Identification of solids collected from the damaged submersible pump section of the methods section of the revised manuscript to explain this to the reader.*

Specific comments

1. This reviewer suggests adding the term 'fractured basalts' to the title given the importance of fractures for CO₂ storage in this scenario. This reviewer suggests the alternative title: *Recirculating water for CO₂-water injection into fractured basalts.* We disagree. *The main breakthrough of this study is the use of recirculating water and*

the success of mineral storage in old and altered basalts. The approach should work just as well in porous basalts. As such we have not altered the manuscript in response to this suggestion.

2. This reviewer suggests changing the second sentence of the abstract for improved clarity. *We agree and have revised this sentence in the revised manuscript to improve its clarity.*
3. This reviewer suggests that desalinization facilities do not produce CO₂ emissions themselves (Scope 1 emissions) on page 2 line 12 of the manuscript and considers removing these words. *Perhaps true in most of the world, but in Saudi Arabia, it is common practice to use oil to run dual electricity- desalination plants, and these are large CO₂ emitters in this region. Thus, the original sentence is correct and we have not change this in the revised manuscript.*
4. This reviewer asks of there more information on the burial and uplift of the region over the past 20 to 30 million years on page 2, line 28. *We agree and have added a sentence and a reference to inform the reader of the past geologic history of the region.*
5. This reviewer asks that we provide the distances of the wells that had no hydraulic connectivity to DW-1 and DW-3 on page 2, line. 31 and asks if the location of the three wells be included in Fig. 1? *In response to this suggestion, we have added a map showing the relative positions of all the wells drilled for this project as an extended data figure and referred to this in the caption of Figure 1.*
6. This reviewer suggests we emphasize the importance of the orientation of the horizontal faults and fractures more strongly in Figure 1 and in text on page 4, lines 3 to 6. *We agree and have added a sentence to the revised manuscript to alert the reader to the existence of sub-horizontal fractures in this system.*
7. This reviewer asks that we describe how we determined DIC concentration on page 4 line 19. *This was described in the methods section of the original manuscript. Consequently, we have not changed the manuscript in response to this comment.*
8. This reviewer asks for insight into why primary plagioclase and pyroxene strongly undersaturated before the commencement of CO₂ injection on page 5, line 9. *We attribute this to the drawdown of fluids from near the surface to the production well. The pumping to the production well lowered the water level in the production well by 150 m. This resulted in the downward migration of cooler, shallower water regions down to the production well, leading these primary phases to be undersaturated in collected production well fluids. We have altered the text of page 5 of the revised manuscript to clarify this to the reader.*
9. This reviewer suggests we rewrite the sentence on page 5, lines 12 – 15 for improved clarity. *We have altered this sentence somewhat in the revised manuscript, but we believe that this was and continues to be clear in the revised manuscript.*
10. This reviewer suggests we consider the role of zeolite formation as secondary minerals on page 5, line 31. *These minerals are undersaturated in all production well fluids so unlikely to form. We have added a sentence to this part of the revised manuscript to alert this to the reader.*
11. This reviewer expresses concern about using the NaF to dissolved carbon ratio to estimate the quantity of the carbon mineralization page 7, lines 1 – 2: This reviewer is concerned that NaF may not be conservative and to provide further insight into the

decline in SF₆ at the production well. *We agree and have added a paragraph to the methods section discussing this issue. Notably more than half the Na-F was recovered during the monitoring period, and modelling suggests more would arrive in the future. Moreover there is no significant difference between the fraction of carbon mineralized by using Na-F and SF₆. This further supports our interpretation of the data. We have noted this last factor in new text added to the methods section.*

12. This reviewer asks why we chose diaspore to fix the concentration of aqueous Al on page 9 line 28. This reviewer suggests using clinochlore or laumontite instead. *Diaspore was chosen as it is observed to precipitate rapidly during experiments and as clinochlore and laumontite are calculated to be undersaturated in compared to diaspore in sampled solutions. We have added text to the methods of the revised manuscript to alert this to the reader.*
13. This reviewer asks that we calculate how much of the initial mass of NaF was recovered at the production well on page 11, lines 18 – 21, and to use this information to comment on the potential loss of Na-F through absorption and diffusion into the matrix. *We agree and have added a paragraph to the methods section providing this number and discussing the implications.*

Referee #3:

1. This reviewer asks us to estimate the change in porosity from the dissolved mineral and precipitated mineral mass balance to derive a crude lifetime and total carbon capacity for the trial system (acknowledging heterogeneity of both precipitation and primary vs secondary porosity response). He continues to ask how might this scale up further and how reliant is this on the secondary porosity? *We agree that these questions are critical to upscaling mineral carbonation to the global scale. These estimations depend on numerous factors including fluid injection rate, the effectiveness of fracking, and the composition of the injected fluid. We have attempted to illuminate these challenges to the reader in a section added to the end of the methods section of the revised manuscript.*
2. This reviewer asks us to estimate the power consumption of recirculation to address the feasibility of this process beyond the water volumes used. *We agree. In a hydrostatic environment, the power consumption of any pumping system is related to the depth of the water table, this indeed takes energy. In contrast the injection itself is gravity driven and does not. This more than makes up for the energy needed to pump water to the surface. We have added text to the end of the methods section to inform this to the reader.*
3. This reviewer asks us to consider what level of conditioning (e.g. fracking) may be required to roll out this type of carbon storage approach and if it is practical. *We have discussed this among ourselves numerous times. It is difficult at present to assess the potential impact of factors such as fracking on the efficiency of mineral storage at present as it has, to our knowledge never been tried in basaltic rocks. In response, we have added a brief discussion on the potential of fracking to the end of the methods section.*

Reviewers Comments

In addition, in your e-mail you proved a second set of review comments from the three original reviewers. These reviewers found that most of their concerns were adequately addressed, by expressed several remaining concerns. Below we repeat only these further concerns and our responses.

Reviewer 1.

This reviewer had one major concern. Specifically, this reviewer finds that our method used to clean the production well sampling bottles be problematic. In the opinion of the reviewer there are various anions and cations both in the original drinking water as well as in the well water that can adsorb to the bottle walls. As a consequence, this reviewer questions the quality of our ICP-OES measurements

We disagree with this suggestion. To support our use of these bottles, we have taken several of these bottles, drained them, and added distilled demineralized water plus the same concentration of acid originally added to the production well water samples. We shook these for 2 days and measured the elemental concentration of the water using the same ICP-OES as used in our study. All measured element concentrations were below the detection limit. We have reported this result on lines 434 to 438 in the methods section of the revised manuscript.

This reviewer asks that we clarify the acidification used for the ICP-OES analysis

We agree that this acidification procedure could be clearer. We have thus revised this description to lines 414 and 415 of the in the methods section of the revised manuscript.

Please also change in Figure captions and Figure labelling as well (e.g. Figure 2). *Though we are not completely clear on what changes are being requested here, we have proofread and correct a small error we found in the caption of Figure 2 and a small labeling error in Figure 3.*

This reviewer suggests we change NaF to Na-F in Extended Data Figure 5.

We agree and have made this change to both extended Data Figure 5 and 6 in the revised manuscript.

This reviewer suggests that it would be beneficial to include the information in the text that the desalination plants in Saudi Arabia are large CO₂ emitters – as it's not clear how the reader is supposed to know that if it's a local specialty.

We agree that the large carbon emissions from desalinization plants are unusual. It seems tangential to add this information to what is now the first paragraph, but the Hamieh reference (reference 13) cited in the sentence described the typical and high carbon emissions from this source. As a result, we have not changed the manuscript in response to this comment.

Reviewer 2

Unresolved Main comments.

1. This reviewer appears to request further discussion on the conservative behavior of the Na-F in the light of the results of Black et al. (2017)

doi.org/10.1016/j.ijggc.2017.10.012.

We acknowledged that Black et al. (2017) reported that they observed substantial non-conservative tracer of Na-F during their study of CO₂ injection at the CO2CRC storage site. During the CO₂ injection considered in their study, Na-F and CO₂ were co-injected together with a number of oxic gases including NO₂ and SO₂. Evidence for redox reactions occurring in the subsurface were reported in that study, including an increase in dissolved aqueous H₂S concentration. In contrast, in the present study CO₂ was injected in without addition of oxic gases, and the system was isolated from the atmosphere to avoid additional of O₂ from the atmosphere. The conservative behavior of the injected Na-F tracer in our study is indicated by the large percentage of tracer recovery (greater than 50%) and that the results generated from Na-F are essentially identical from that observed using SF₆. We believe that the difference between the behavior of the Na-F tracer in our study from that of Black et al (2017) stems from the presence of injected dissolved oxic gases, as the Na-F tracer is known to oxidize in the presence of a number of oxidizing agents. In fact, this property is widely used in biochemistry to characterize metabolic processes. Perhaps more significantly, if Na-F was sorbed as observed in Black et al (2017) study, the calculated mineral percent calculated in our study would be an underestimate, and the true mineralization percent would have been greater than we report in the manuscript. We have explained this to the reader in a new paragraph added the end of the 'From Sodium Fluorscein' section of the further revised manuscript.

2. This reviewer asks again why SF₆ concentrations decline over time at the production well given it is continuously added to the injection water and asks what is the sink for SF₆?

I think that this comment arises as we were unclear in the text about the variation of the SF₆ concentration over time in the original text. The variation of this concentration first maximizes shortly after the addition of SF₆ to the recirculating fluids was stopped on 7 July 2023, then it decreased in the sampled production well fluids afterwards. This is thus not inconsistent with the behavior anticipated by this reviewer. To clarify this behavior, we have added additional text on lines 599 to 601 of the 'from SF₆' section of the methods section, and have re-formatted the axis of Figure 5b and c to better show the reader when CO₂ and SF₆ were added to the subsurface system.

4. This reviewer asks again for us to comment on the potential effects of diffusion from the fractures to the adjacent matrix rock for components enriched in the injection well (DIC, H⁺, liquid tracers). Specifically, this reviewer is concerned that the loss of dissolved CO₂ is (to some degree) controlled by the transport away from the main fluid flow pathways and not by carbonate precipitation.

Diffusion of aqueous components into the rock matrix would not alter our estimates of carbon mineralization fraction as long as the injected tracers and CO₂ diffused into the rock matrix at the same rates. This is almost certainly the case as the tracers, CO₂ and hydrogen are all aqueous dissolved species. We have added a sentence to alert this to the reader on lines 543 and 544 of the the section 'From Sodium Fluorscein' section of the further revised manuscript.

Unresolved Specific comments:

1. This reviewer suggested adding we add the term 'fractured basalts' to the title given the importance of fractures for CO₂ storage in this scenario. Specifically, this reviewer suggests we use the title: Recirculating water for CO₂-water injection into fractured basalts.

We agree with the reviewer that adding the word basalts to the title would be informative, and in response we have added this word. We disagree that the word fractured is necessary as the described process would work equally well in porous basalts, and that our system contains some bulk porosity. In addition, the addition of the word fractures, would increase the title length beyond the normal accepted title length, so this word was not added.

11. This reviewer again expresses concern about using the Na-F to dissolved carbon ratio to estimate the quantity of the carbon mineralization page 7, lines 1 – 2 of the original manuscript.

We have addressed this comment above in response to general comment 1.